# Unraveling the deep learning gearbox in optical coherence tomography image segmentation towards explainable artificial intelligence

Peter M. Maloca [1,2,3,4✉], Philipp L. Müller [4,5], Aaron Y. Lee[6,7,8], Adnan Tufail[4], Konstantinos Balaskas[4,9], Stephanie Niklaus[10], Pascal Kaiser[11], Susanne Suter [11,12], Javier Zarranz-Ventura [13], Catherine Egan[4], Hendrik P. N. Scholl[1,3], Tobias K. Schnitzer[10], Thomas Singer[10], Pascal W. Hasler[2,3] & Nora Denk[3,10]

Machine learning has greatly facilitated the analysis of medical data, while the internal operations usually remain intransparent. To better comprehend these opaque procedures, a convolutional neural network for optical coherence tomography image segmentation was enhanced with a Traceable Relevance Explainability (T-REX) technique. The proposed application was based on three components: ground truth generation by multiple graders, calculation of Hamming distances among graders and the machine learning algorithm, as well as a smart data visualization ('neural recording'). An overall average variability of 1.75% between the human graders and the algorithm was found, slightly minor to 2.02% among human graders. The ambiguity in ground truth had noteworthy impact on machine learning results, which could be visualized. The convolutional neural network balanced between graders and allowed for modifiable predictions dependent on the compartment. Using the proposed T-REX setup, machine learning processes could be rendered more transparent and understandable, possibly leading to optimized applications.

[1] Institute of Molecular and Clinical Ophthalmology Basel (IOB), Basel, Switzerland. [2] OCTlab, Department of Ophthalmology, University Hospital Basel, Basel, Switzerland. [3] Department of Ophthalmology, University of Basel, Basel, Switzerland. [4] Moorfields Eye Hospital NHS Foundation Trust, London, UK. [5] Department of Ophthalmology, University of Bonn, Bonn, Germany. [6] Department of Ophthalmology, Puget Sound Veteran Affairs, Seattle, WA, USA. [7] eScience Institute, University of Washington, Seattle, WA, USA. [8] Department of Ophthalmology, University of Washington, Seattle, WA, USA. [9] Moorfields Ophthalmic Reading Centre, London, UK. [10] Pharma Research and Early Development (pRED), Pharmaceutical Sciences (PS), Roche, Innovation Center Basel, Basel, Switzerland. [11] Supercomputing Systems, Zurich, Switzerland. [12] Zurich University of Applied Sciences, Waedenswil, Switzerland. [13] Institut Clínic d'Oftalmologia, Hospital Clínic de Barcelona, Barcelona, Spain. ✉email: peter.maloca@iob.ch

Machine learning (ML) algorithms learn to perform a specific task without being explicitly programmed to use conventional logic rules[1,2]. In artificial intelligence (AI), the technique of deep learning (DL) that utilizes multi-layered artificial neural networks or convolutional neural networks (CNNs) for image segmentation is considered to be one of the most promising tools in medicine[3–6].

Giving that the modern ophthalmic diagnostic increasingly relies on imaging, especially the use of optical coherence tomography (OCT)[7] and image analysis, the field of ophthalmology is particularly suited to be focused on by ML applications. OCT is a non-invasive imaging technology that utilizes low-coherence laser light to produce cross-sectional images in biological tissues[8–10]. There have been successful reports with regard to implementation of ML-based analysis of OCT data and its diagnostic accuracy for neovascular age-related macular degeneration[11–13], diabetic retinopathy[14–18], or retinal vein occlusion[19–21], among others[22]. Machine learning increased OCT information throughput and showed similar performance as human graders in annotation of complex OCT images[23–26]. Recently, the OCT image assessment capabilities of ML were complemented by the introduction of a three-dimensional CNN and the ML approach showed a performance in making a referral recommendation that was at least as good as human experts[23]. Despite the potential of ML applications[26–37], it is not yet useful in clinical routine. As hitherto applications take place in a relatively narrow and predictive environment[38,39] under tightly defined rules and conditions[40–43], the flexibility and inferential reasoning in unforeseen and critical situations is still unknown. Thus, a better understanding of possible challenges[44] when deploying ML will be helpful in order to evaluate the field of application, limitations, and the reliability[37,45].

The discrepancy between how a computer works and how humans think is known as the "black box problem"[46]: in communication technology and engineering language a system is usually considered as a "black box" that features an input and output path and shows a particular or at least statistically definable sort of operation. However, such a solution either is not specified in all details or cannot be visualized, so that its mode of working remains unidentified or hidden or in a way that is not (yet) comprehensible to humans. This can cause a major issue due to the frequent incomplete knowledge and interpretability of the algorithm's internal workings, in particular, for DL models[6,47].

Investigating the AI black box[48–50] has become known as explainable AI (XAI)[51–53], which provides tools that reveal the AI decisions. The call for transparency of AI models is especially high in medicine, where uncertainty, ambiguity, and the unknown are inherent to the discipline. XAI distinguishes taxonomies[53,54] such as *understanding* referring to comprehend the inner mechanisms of an AI model; *explaining* revealing "why" a machine technically decided for an outcome based on a collection of features that contributed to the AI decision; and *interpreting* mapping the abstract (and technical) XAI concepts to a human understandable format. While humans often cannot explain the reasoning behind a decision, understanding an AI model's decision process will provide confidence and acceptance of the machine.

Further knowledge is achieved by *causability* approaches[55], which measure the quality of explanations produced by XAI techniques in the human intelligence domain, e.g., with the System Causability Scale[56]. Post hoc XAI techniques such as LIME[57], BETA[58], GradCAM[59–61], LRP[62,63], Deep Taylor decomposition[64], or TGAV[65] highlight and visualize regions of the input data that lead to relevant prediction decisions after the neural network training process (post hoc).

In contrast to the previously mentioned reports, in this work, we enhance ML explainability with a post hoc XAI technique, which we suggest terming Traceable Relevance Explainability (T-REX) of graders. Supervised machine learning requires labeled ground truth data, which is usually annotated by humans in a highly time-consuming process. However, in some domains there is no absolute consensus on what the true ground truth labels should be. This is particularly true in medical imaging. Different experts might judge the same medical images slightly differently and come to different conclusions as to where for example the borders of certain medical structures should be marked[24]. In some cases, the ambiguity might be irresolvable, i.e. there is no unambiguous ground truth criterion, since it is impossible to determine the exact location of these structures without applying invasive and destructive procedures to the patient. In these cases, we might want to better understand how an ML algorithm reacts to ambiguity in the ground truth data.

To shed light on this issue, we trained a CNN from ambiguous ground truth consisting of labels from three human graders who acted as three CNN "teachers". In particular, we applied the proposed XAI method T-REX to automatic OCT image segmentation. Using our XAI technique T-REX, we propose to post hoc record and evaluate the variability of the CNN predictions relative to the human graders by using Hamming distances and then visualizing and analyzing the measured variability with heatmaps and multi-dimensional scaling (MDS) plots. Using this approach, we highlight similarities of the trained CNN to the varying human graders (or "teachers") that the CNN has learned from which could be referred to as a kind of ML neural recording. We assessed and visualized how a CNN learned from ambiguous ground truth data and independently positioned itself between the human graders with respect to how the humans labeled the ground truth.

In summary, the specific contributions of the study are as follows:

DL model for species-specific retina OCT image segmentation, the first time for non-human primates, i.e., cynomolgus monkeys.

CNN training from ambiguous OCT ground truth labeled by several independent human expert graders.

Proposing the XAI technique, termed T-REX, for the recording and visualization of the predictive performance of a CNN with respect to ambiguous ground truth generated by multiple graders.

Applying T-REX to reveal that a CNN trained on ambiguous ground truth learned a form of consensus among the human graders, which is eye compartment-specific (vitreous, retina, choroid, sclera).

## Results

The graders were 42 years old on average (range from 35 to 53 years). Average medical work experience was 14 years. Grader 1 was an experienced male retina specialist with a work experience in ophthalmology of 25 years. Grader 2 and 3 were females (one veterinary physician and one neuroscientist) with a work experience in ophthalmology of 6 and 13 years. Expertise in OCT imaging was 22, 6, and 2 years, respectively, for grader 1, 2, and 3, respectively.

The proposed CNN showed a successful implementation of automated segmentation of retinal OCT images in animals. The performance results (i.e., Hamming distance) of the three human graders and the proposed CNN with respect to the test set (200 B-scans of eight eyes) are summarized in Fig. 1.

With respect to the reliability of confining the correct compartments, the variability of the vitreous and the retina compartments was smallest with values below 0.7% for all gradings. The distinctively higher overall variability, presented above, might therefore originate from the delimitation of the choroid and sclera compartments, which are separated by the annotated choroidal sclera interface (CSI) line (Figs. 2 and 3).

Interestingly, the relative variability (i.e. the level of variability of one pair in dependence on the overall level of pairwise variability) partly changed with the compartment (Fig. 3): While g1 and g3 always showed high and g2 and g3 low relative inter-grader variability, the relative inter-grader variability of g1 and g2 was high in the vitreous and the retina, but low in the choroid and the sclera. While the relative variability of g2 and CNN was consistently in the lower half for each compartment, the relative variability of g1 and CNN and g3 and CNN differed in dependence of the compartment. Initial experiments on the smaller ground truth set based on 800 B-scans yielded very similar results indicating reproducibility of the results.

The MDS plots in Fig. 4 show mappings of the three human graders (g1, g2, and g3) and the CNN as points into a Cartesian coordinate system such that the physical distance between the points corresponds to the mean Hamming distance and is preserved as well as possible. If all markers were on top of each other in a MDS plot, all human graders and the CNN would have made identical gradings; hence, the farther away the markers are, the more differences exist between two gradings. The distance from the CNN (red dot) to the human markers g1, g2, and g3 (black triangles) visualizes the similarities between the CNN predictions and the human gradings and correspond to the above-described relationships. The overall CNN training variability is shown in Fig. 4a, where it is visible that the CNN is similarly close to g1 and g2. For the vitreous (Fig. 4b), the CNN lies similarly close to g2 and g3; for the retina (Fig. 4c) the CNN lies closest to g2, which is not far from g3; for the choroid and sclera (Fig. 4d, e), the CNN lies closest to g1, which is not far from g2.

To better understand how the recorded predictive performance of the CNN relates to the ambiguity in the ground truth data, individual Hamming distances per B-scan are shown in Fig. 5, shifting the attention from the mean performance of the CNN across the whole test set to a more thorough investigation on a per B-scan level. Investigating the mean predictive performance of a CNN does not trivially allow deducing general properties about a CNN's predicting behavior with respect to ambiguity in ground truth data. This is only possible by an analysis of the recorded CNN predictions per B-scan: at a glance, it is visualized that the CNN shows little variation to human graders over each

single B-scan. However, it is recorded and depicted that among human graders the variation per image can be much higher which is not obvious when considering only mean Hamming distances: The plot shows that the human graders g1 and g2 labeled relatively similarly and that g2 and g3 labeled relatively similarly as well (Hamming distances highlighted by predominantly green color). On the other hand, the labels between g1 and g3 are more different (Hamming distances highlighted by predominantly yellow color). CNN predictions generally represent some sort of average among the three human graders. The mean of the three grader-CNN Hamming distances was smaller than the mean of the three inter-grader Hamming distances in 78.5% of the B-scans. The CNN predictions are usually closer to g1 and g2 (Hamming distances highlighted by predominantly green color) than to g3 (small to moderate Hamming distances are highlighted by green and yellow colors).

As specified below, the statistical permutation tests revealed that the mean human inter-grader Hamming distances were significantly larger than the mean Hamming distances between the humans and the CNN across all compartments and for the compartments vitreous, choroid, and sclera separately. The recovered $p$ values were all $2e-5$ with a 99% confidence interval of $(0, 1e-4)$ for the $p$ values. This is, of all the permutations drawn, not a single time did the mean Hamming distance between humans and the CNN exceed the mean human inter-grader Hamming distance. On the other hand, for the retina compartment the mean inter-grader Hamming distance was remarkably smaller than the mean Hamming distance between the humans and the CNN. The recovered $p$ value was again $2e-5$ with a 99% confidence interval for the $p$ value of $(0, 1e-4)$.

## Discussion

In the past few years, OCT has been rapidly implemented into diagnosis[7,13,66] and monitoring of retinal diseases[67,68]. Currently, such measurements are widely used in humans[69–71], but not routinely employed in animals (Fig. 6). Hence, this is the first report of an automated DL segmentation of vitreo-retinal and choroidal compartments in healthy cynomolgus monkeys, a species commonly used as animal models of human disease as well as for safety assessment in preclinical trials. The translation of a previously developed and reproducible ML framework in humans[24] to animals was successful. This suggests that the basic DL framework was also applicable to animals after the ML was specifically adjusted and trained on animal data.

While ML enhanced the discovery of complicated patterns in OCT data and showed similar performance to humans[37], there is still an unmet need for a better understanding on how ML exactly learns[72–74]. Typically, data are inserted into an ML environment and the results produced on the other side are often associated with a great degree of uncertainty about what is happening in between. This is referred to as ML black box (BBX). Clarifying this black box entails creating a comprehensive ML approach ideally designated for the human cognitive scale.

Thus, to solve part of this black box issue and to foster transparency, we propose an ML display concept which is designated as T-REX technique. T-REX is based on three main components or gears: ground truth generation by several independent graders, computation of Hamming distances between all graders and the machine's predictions, and a sophisticated data visualization which is termed as neural recording (NR) of machine learning. In analogy to a mechanical gearbox, consisting of an arrangement of machine parts with various interconnected gears, we understand an ML gearbox to be composed of fine-tuned software elements which, when properly linked, should provide insight into the inner workings of the entire machine. In

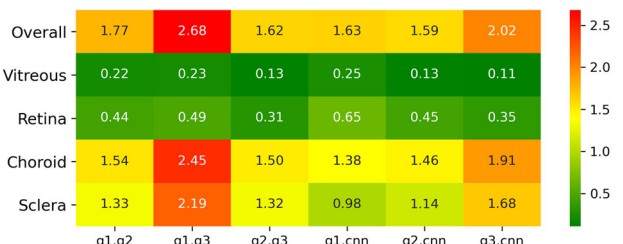

| | g1,g2 | g1,g3 | g2,g3 | g1,cnn | g2,cnn | g3,cnn |
|---|---|---|---|---|---|---|
| Overall | 1.77 | 2.68 | 1.62 | 1.63 | 1.59 | 2.02 |
| Vitreous | 0.22 | 0.23 | 0.13 | 0.25 | 0.13 | 0.11 |
| Retina | 0.44 | 0.49 | 0.31 | 0.65 | 0.45 | 0.35 |
| Choroid | 1.54 | 2.45 | 1.50 | 1.38 | 1.46 | 1.91 |
| Sclera | 1.33 | 2.19 | 1.32 | 0.98 | 1.14 | 1.68 |

**Fig. 1 Heatmap table of the test set grading variability.** The deviations are shown in percentage (%) between human graders (g1–g3) and the CNN with regard to the ocular compartments. The percentages are equivalent to the Hamming distance and represent the difference in labeled pixels between two gradings. In particular, the inter-human grader variability and the variability between the human graders and the CNN are shown. The overall average inter-human grader variability between (g1, g2), (g1, g3) and (g2, g3) was 2.02% and 1.75% between the CNN and the human graders, i.e., (g1, cnn), (g2, cnn), and (g3, cnn). Overall, the results of grader g2 and g3 were most similar, followed by the differences in the gradings of g2 and g1. The segmentation of grader g1 and g3 differed most. When comparing the segmentations of the human graders and the CNN, the Hamming distance increased from CNN compared to g2, to CNN compared to g1 to CNN compared to g3. Regarding individual compartments, agreement in vitreous and retina are higher than in sclera and choroid.

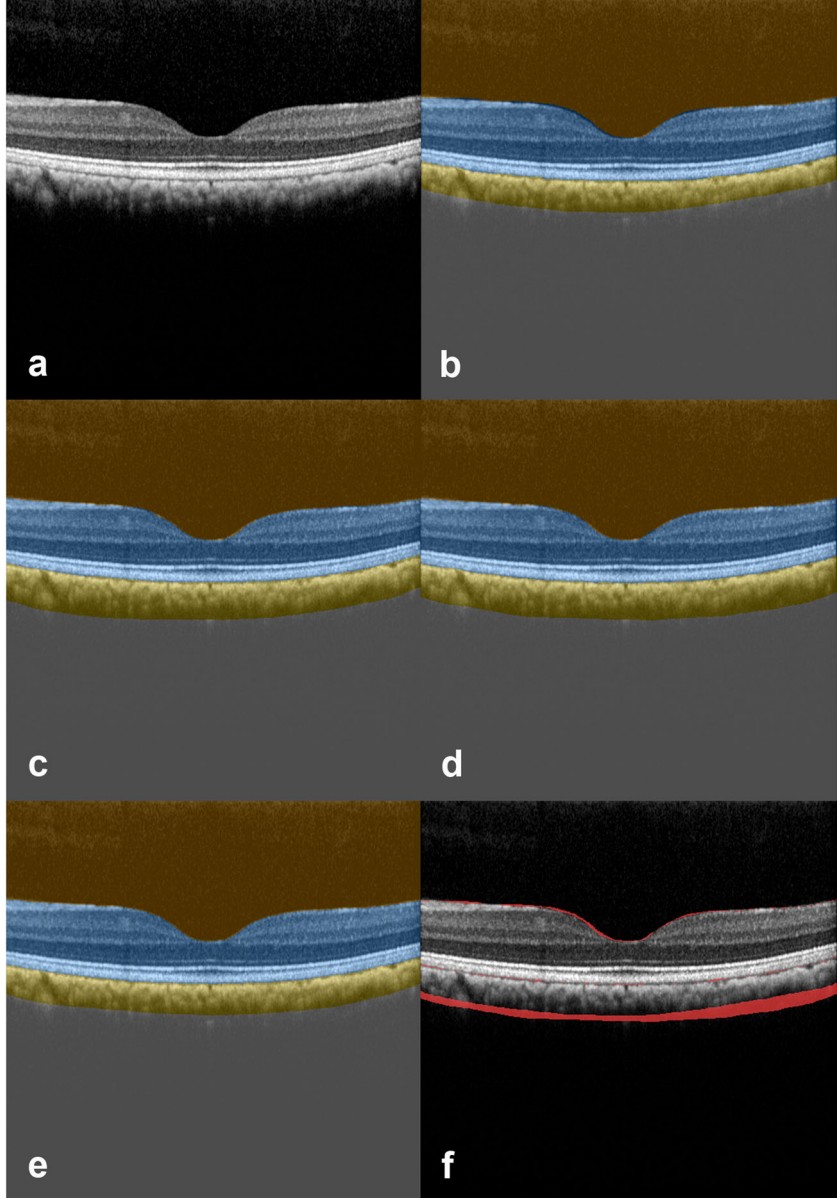

**Fig. 2 Optical coherence tomography image segmentation.** Visualization of an exemplary test set result from a cynomolgus monkey's left macula. OCT B-scan image **a**, with corresponding image segmentation results derived from three human graders: **b** g1, **c** g2, **d** g3, and **e** the CNN. There was an excellent inter-grader and grader-CNN agreement for the segmentation (vitreous, orange; retina, blue; choroid, yellow; sclera, gray). The highest variability was found in the delineation of the choroidal sclera interface separating choroid and sclera. **f** highlights pixels in red color that were labeled differently by g1 and g3. The number of pixels in red color divided by the total number of pixels in the image is the Hamming distance between g1 and g3.

this sense, the ML learning process would receive a better and appropriate appreciation to determine what characteristic data the algorithm uses to make decisions. Given the overlap between neuroscience and ML, we understand by the notion of NR the registration and visual display of the predictive performance of a machine learning algorithm and human graders related to the ambiguity in the ground truth data, so that the values are presented in a comprehensive way to ML experts but are also suitable for people with a lower level of ML expertise. This is even more important to reach a larger audience so that researchers outside the ML domain who are less familiar with ML complexities can obtain a more straightforward approach to the findings.

To facilitate understanding of the Hamming distance values, visual representations of the data were conducted using MDS plots (Fig. 4) and a heatmap plot (Fig. 5). With regard to the aforementioned, this form of neural recording enables data scientists, regulators, and end users like medical doctors to better understand the impact each human grader had on the predictive performance of a trained machine learning model and thereby enhancing the understanding of a machine's decision process. Therefore, an interesting added value of this study is that it is now possible to develop a more detailed understanding of what a CNN values in learning from each single OCT image annotation. Thus, with regard to the decision-making process of a CNN, the depth of the level of detail considered; in short, the ML decision granularity was increased. The proposed T-REX methodology showed which part of the ground truth was more important: generally, graders 1 and 2 were more relevant than grader 3 because the CNN was almost always located closer to g1 and g2 (Figs. 4 and 5). Thus, g1 and g2 seem to have influenced the CNN more during learning.

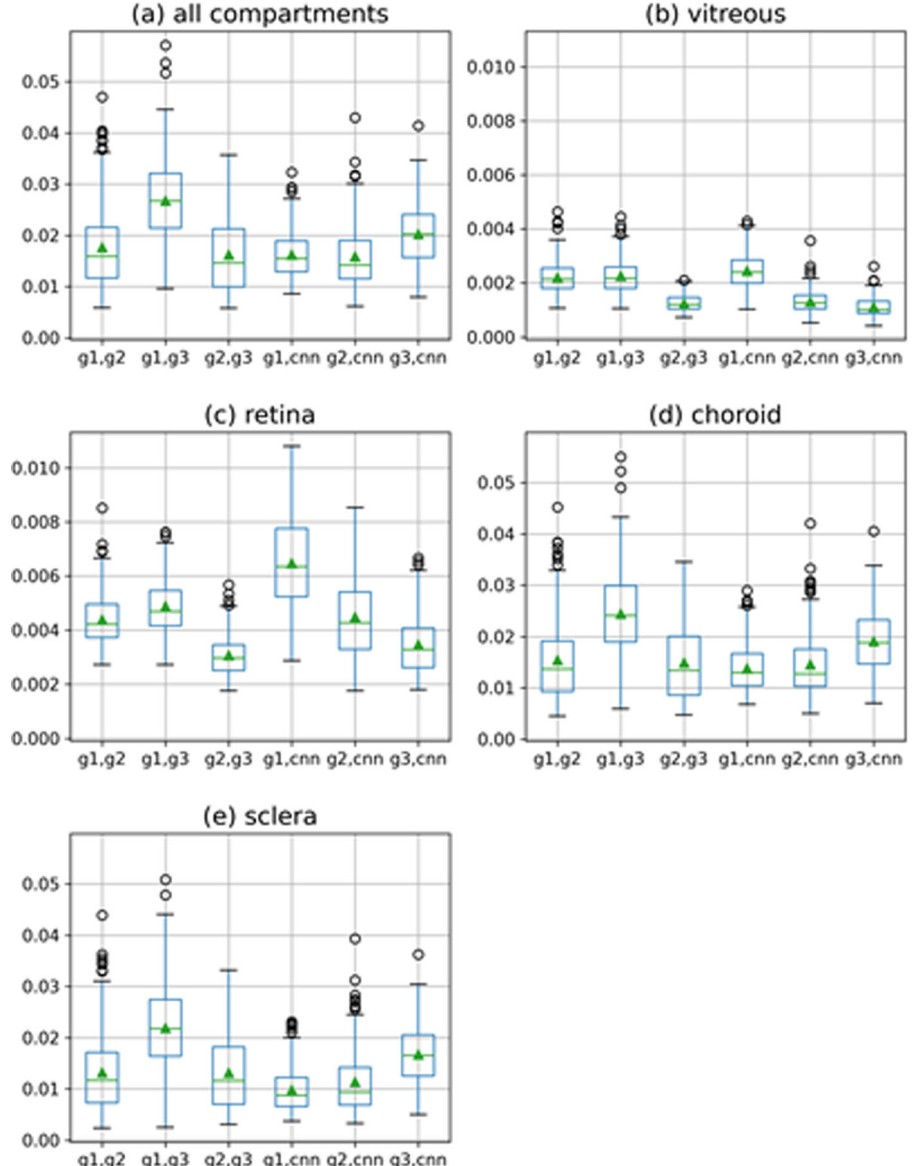

**Fig. 3 Boxplots of Hamming distances as the percentage of pixels that were labeled differently.** The distribution of variability is shown between segmentations of the human graders (g1–g3) and the CNN predictions. Boxplots show pairwise comparisons of g1 with g2, g1 with g3, g2 with g3, g1 with CNN, g2 with CNN, and g3 with CNN. **a** shows the variability over all compartments, **b** for the vitreous compartment, **c** for the retina compartment, **d** for the choroid compartment, and **e** for the sclera compartment. Rectangular boxes in boxplots represent interquartile ranges (IQR) and extend from quartile Q1 to quartile Q3 with green lines and green triangles indicating median (Q2) and mean, respectively. Upper whiskers extend to the last datum which is less than Q3 + 1.5 × IQR. Lower whiskers extend to the first datum which is greater than Q1 − 1.5 × IQR. Data beyond whiskers are considered as outliers and are plotted as individual circles. Note: the retina and vitreous axes are zoomed in to demonstrate the compartment-specific variability. Overall and for each compartment but the retina, the total inter-grader difference was larger than the difference between human graders and the CNN.

Interestingly, these findings also correlate with the level of OCT expertise. Grader g1 and g2 have a higher level of expertise in OCT imaging than grader g3. So, it could be assumed that g1 and g2 generated more consistent annotations, which could have drawn the CNN predictions closer to them than to grader g3. This adaptive performance can be assumed to be directed, i.e., it seems to be "aim-oriented". Such a mode of behavior is usually attributed to the term "intelligence".

Overall, a good predictive performance was observed: The minor overall average per-pixel variability between the trained CNN and the human graders (1.75%) was notably lower than the inter-human variability (2.02%). The gross range of variability was congruent to previous reports[75,76]. Our results unraveled the CNN's problem-solving skills and behavior as a form of learning

a kind of robust average among all the human graders. This fact further supports the utilization of DL-based tools for the task of image segmentation, especially as the CNN performs the same task repeatedly producing the same output—independent from any physical or mental state compared to humans.

For comparing our results of the OCT segmentation to previous works, we put our results in the context of an analogous study in humans[24]. The study design differs, but still, the comparison gives insight into the CNN performance. In the study with humans[24], a CNN was trained based on only one experienced grader, and verified with multiple human graders at three points in time. In contrast, in this study, the CNN was trained and verified with the same three human graders that labeled the images at one single point in time. In the human study, the overall inter-human

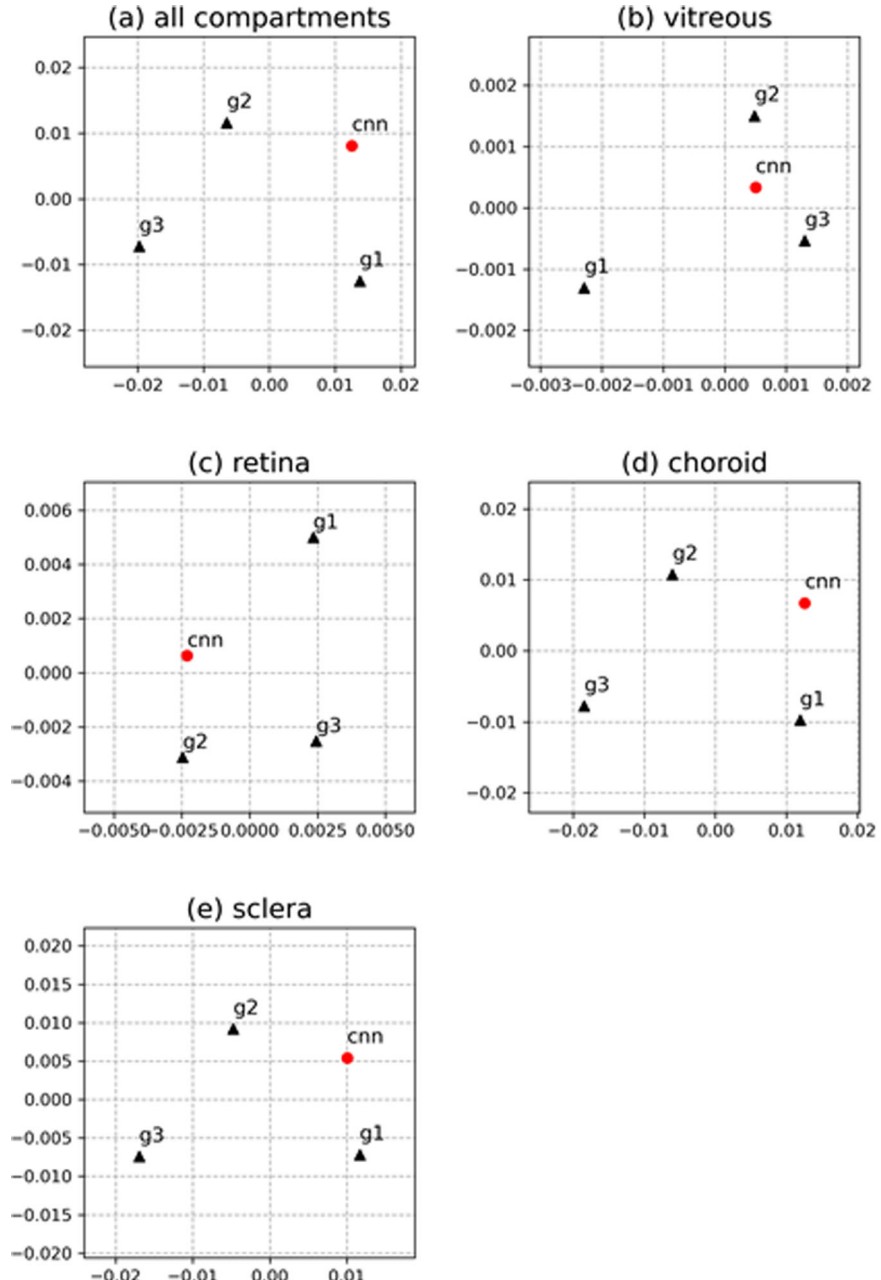

**Fig. 4 Multidimensional scaling plots of mean Hamming distance.** Results between labels of grader 1 (g1), grader 2 (g2), grader 3 (g3), and CNN predictions are depicted. The axes have no unit and represent Hamming distances. Multidimensional scaling places g1, g2, g3, and CNN in a two-dimensional coordinate system such that distances between g1, g2, g3, and CNN correspond to the mean Hamming distances as accurately as possible. Multidimensional scaling plots are shown for **a** all compartments, **b** vitreous compartment, **c** retina compartment, **d** choroid compartment, and **e** sclera compartment. g1, g2, and g3 are shown as black triangles. CNN is shown as a red circle. Depending on the compartment, the CNN showed a different averaging behavior with respect to the human graders.

variability was 2.3%, and the overall human–CNN variability was 2.0%, while the variability of three runs of the ground truth grader with the CNN was 1.6%. The range of variability was also congruent to previous reports[75,76]. As these numbers are higher than in the monkey study presented here, the actual improvements in the study design consequently might increase the performance of the presented CNN. The balancing behavior pattern during the CNN prediction, as unveiled with T-REX, reveals that in such an ML study, it is advisable to train the CNN with several graders—not just with a single gold standard expert. The proposed study design makes a CNN more robust and inherently includes an external validation[37].

By analyzing the ML Hamming distance patterns, evidence has not only been found to support an actively balanced type of computational ML regime that can underlie any ML procedure. A similar performance was also shown in cortical circuits although of course artificial neural networks represent very rough simplifications of brain functions[77]. Dependent on the compartment, i.e., characteristic data label, the CNN judges the importance of the labels of the three human graders during training differently. For the vitreous and retina compartments, g2 and g3 produced labels relatively similarly, and g1 produced labels relatively differently. During training the CNN seems to pay more attention to the labels of the two graders who labeled similarly since the mean

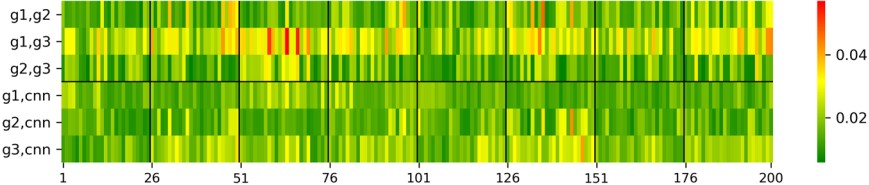

**Fig. 5 Illustration of neural recording results.** Heatmap of Hamming distances across all compartments for each individual B-scan (test set of 200 B-scans from 8 eyes). Hamming distances are shown between g1 and g2, g1 and g3, g2 and g3, g1 and CNN, g2 and CNN, and g3 and CNN. Green and red colors indicate small and large Hamming distances, respectively. Each eye contributed 25 B-scans. Thin vertical lines in black color separate eyes and the horizontal black line separates inter-human and machine versus human recordings, respectively. The ordering of B-scans within each eye was constant and no systemic difference was found regarding the location of the B-scan within the retina, i.e. whether it was a peripheral scan or a scan of the macula. There was very little variation across all individual B-scans (mostly greenish colors). Nevertheless, it is interesting that between the graders a few B-scans are shown in red, whereas compared to CNN no images were marked in red.

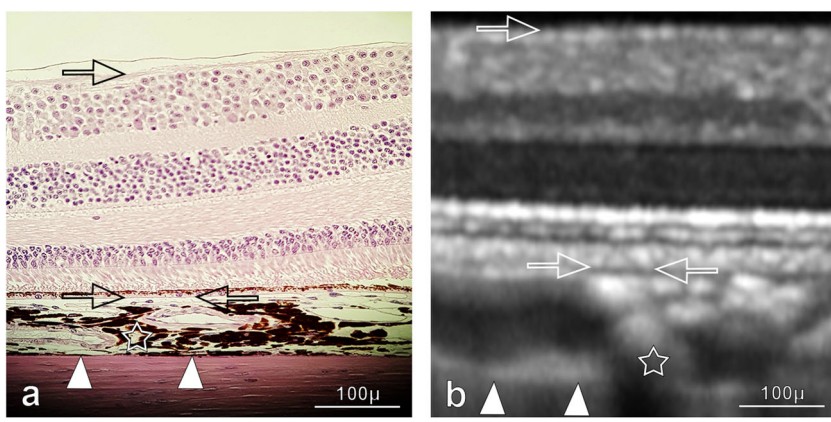

**Fig. 6 Multimodal imaging of retinochoroidal zones in a healthy cynomolgus monkey's eye. a** A high-resolution histological hematoxylin and eosin staining of a paraffin-embedded cross-section of a normal cynomolgus monkey's eye. **b** Corresponding OCT B-scan from another cynomolgus monkey's eye. Illustrated in both images are the vitreous-retina border (ILM, single arrow), internal part of choriocapillaris (CCi, double arrows), and choroid–sclera interface (CSI, arrow heads).

Hamming distance across the test set of 200 B-scans is closer to g2 and g3 than to g1 (Fig. 1). On the other hand, for the choroid and the sclera compartments, g1 and g2 labeled relatively similarly and g2 and g3 labeled relatively similarly. But the labels of g1 and g3 were relatively different. In this case, the CNN learned to make predictions that are closer to g1 and g2 than to g3. This behavior can be compared to a gear shift. Depending on the compartment, i.e. the data label, the CNN applies a different learning strategy with respect to the ground truth data. This demonstrates the importance of employing multiple independent graders for CNN training. Although this is a well-known and expected phenomenon in machine learning, it is nevertheless remarkable that this circumstance has become visually representable with this work and perceptible in such a way.

T-REX, our proposed XAI approach, can be helpful to narrow down the numerous possibilities for the development and enhancement of artificially generated knowledge; for example, to select which grader provides the best opportunities for ML development or which intentional manipulations induce a deterioration in performance[73]. In particular, our T-REX analysis showed that it is necessary to study not only on the mean predictive behavior of the CNN but also to consider individual predictions on a deeper data level (e.g. each single B-scan) to transform machine learning into valuable learning. Considering only the mean predictive performance could be misleading since it would be possible for a CNN to predict certain images like human grader 1 and others like human grader 2 or 3. Analyzing the predictive performance of a CNN on individual images allows to make more precise statements about the factors that impact the

learning process from ambiguous ground truth data. In general, this will enable the targeted manipulation of the ML framework in the future to document and display performance to objectively benchmark ML models and ground truth data against each other and thus improve and accelerate development. If it is better understood how the machine works, then it will also be possible to work out a set of correct premises to guide the deep neural networks in their learning and to facilitate robustness and generalization.

In order to be able to compare the ML models of different research groups, it would be ideal if they would make not only their code but also their data publicly available. Data sharing is usually restricted due to privacy of health data or data with commercial or intellectual property sensitivity. Therefore, "ML black data" exists beside ML black boxes. T-REX would be an interesting option here to generate indirect clues about the characteristics of such restricted data used so that third parties could better understand and verify the claims made.

Compared to other reports using CNNs a limit of this study could be the relatively low number of annotated ground truth data. However, the average Hamming distance between the human graders and the CNN was 0.0175 corresponding to 1.75% of pixels being labeled differently by the human graders and the CNN, respectively. This high predictive performance of the CNN was confirmed when training on the smaller ground truth data set of 800 B-Scans, which yielded similar results. This indicates that the ground truth size of 900 B-scans was sufficient to sustain the claims proposed in this study. However, it can be speculated that an even higher number of ground truth data could further

improve the results. Nevertheless, the annotation of ground truth data by humans is a very time-consuming process and the current study setup appears to be an acceptable compromise between human effort and CNN predictive performance, particularly considering that the development of ML algorithms often aims at reducing the human workload.

The image quality could also have impacted the results, especially in intensely pigmented eyes due to signal loss. Moreover, there are other possible score systems than the Hamming distance and the Hamming distance does not consider how large the compartments are. In certain situations, e.g., myopia, the choroid can be much thinner than the retina, which could possibly lead to a larger difference, but that was not the scope of this study.

It is worth noting that the individual elements used in this study, i.e. U-Net, Hamming distance, MDS, and heatmap plots, might not be considered as a methodological novelty on themselves. However, the scientific originality of our work can be viewed as a unique combination of pre-existing components[78] or as a permutation of new and old information[79]: T-REX and its associated scientific discoveries in this study provide subsequent studies with a distinctive technique and a combination of knowledge not available from previous reports. In short, the appropriate conceptualization of the mentioned ML elements into the proposed framework improved the understanding of the interface between automatic computing and life sciences and therefore represents nevertheless a specific degree of originality.

Above all and despite all limits, in medicine, physicians will only use an AI system for diagnosis and monitoring of diseases if they can understand and comprehend the internal AI processing. More importantly, physicians will only make a clinical decision based on a recommendation of such an AI system if they can fully identify themselves with the AI. A subset of XAI methods aims at revealing post hoc insights into "why" a machine has taken a certain decision. While well-known post hoc approaches such as LRP or GradCAM visualize relevant regions in the input data, T-REX, our proposed XAI method, visualized and evaluated similarities between the CNN predictions and the labels of different humans that the CNN has learned from. Therefore, this study contributes to a better explainability in the application of AI, such that a resulting DL model can be better appreciated. T-REX can provide a rigorous evaluation and re-calibration tool to incorporate new DL standards. In a more general sense, it can increase the quality of explanations that are based on DL systems, which increases causability[55]. This in turn can promote safety for doctors and patients. Accordingly, the proposed post hoc XAI approach T-REX is expected to enable data scientists to model more transparent DL systems. In return, this leads to further insights into trained DL models by physicians, which utilize DL for data-supported clinical decisions.

The proposed method T-REX is not limited to semantic image segmentation in ophthalmology. In fact, it can be applied to improve the understanding of any machine learning algorithm that learns from ambiguous ground truth data. For example, T-REX could be used in the application of uncovering biases of ML prediction models in digital histopathology not only with respect to data set biases but also with respect to varying opinions of experts labeling the histopathology images[80]. In applications, where supervised ML decision models are trained to detect diseases such as Covid-19 (ref. [81]) and experts still need to explore and agree upon specificities of the particular disease, T-REX would be helpful to visualize the ambiguity of the experts' opinions, i.e., labels. Hence, T-REX might be especially important if the ambiguity is irresolvable meaning that domain experts disagree about the true labels, but the differences cannot be eliminated in a straightforward way. In many medical applications, the true labels cannot be verified because applying invasive

procedures to patients is impossible. Therefore, methods such as T-REX, which highlight the results of the model training from ambiguous ground truth, help to improve the understanding of the objectivity of a trained model and can lead to a reduction of bias in the ground truth.

In a wider context, T-REX might yield insights into how AI algorithms make decision under uncertainty, a process very familiar to humans but so far less understood in the field of AI.

## Methods

**Animals and husbandry.** Retrospective data from 44 healthy and untreated cynomolgus monkeys (17 females, 27 males) of Mauritian genetic background with an age range of 30–50 months and weight ranging from 2.5 to 5.5 kg were used. The use and care of the animals was carried out according to the guidelines of the US National Research Council or the Canadian Council on Animal Care Studies and all procedures complied with all relevant ethical regulations. Inclusion criteria were as follows: cynomolgus monkeys' retinas showing a healthy and a complete display on the OCT B-scans of all four compartments (vitreous, retina, choroid, and sclera), and image quality of at least 25 on an arbitrary unit as indicated by the manufacturer's software. The exclusion criteria were any type of pathology of the retinal layers or choroid recognizable by OCT.

**OCT imagery.** OCT scans were recorded with the Spectralis HRA + OCT imagery platform (Heidelberg Engineering, Heidelberg, Germany) centering on the macula. Each OCT scan was exported to a stack of 25 B-scans with the automatic averaging and tracking feature. B-scans covered a horizontal and vertical length of 5.12 and 4.96 mm, respectively. Distances between consecutive B-scans are fixed and range from 4.8 to 5.4 μm among OCT scans. All B-scans were exported as 24-bit grayscale images with a spatial resolution of 512 × 496 pixels. Each B-scan depicts vitreous, retina, choroid, and sclera compartments. For the training and testing of the CNN, we randomly included the full OCT B-scan stack of either the left or the right eye of one individual.

**Human grading of OCT images.** To train and test the CNN, three physicians manually graded a selection of OCT B-scans. Grader 1 is a retina expert (ophthalmologist); grader 2 and 3 are a veterinarian and a biologist, respectively, working with OCT in preclinical research on a daily basis. The graders generated semantic segmentation maps, i.e., the pixel-wise label annotations, of the vitreous, retina, choroid, and sclera compartments. For each B-scan, the graders manually drew three lines defining the four compartments.

The vitreous compartment was defined as the cavity above the very innermost segmentation line, the hyperreflective region of the internal limiting membrane (ILM, Fig. 6). The retina compartment was outlined by the ILM line and the line placed exactly above the hyporeflective zone referred to the choriocapillaris (CC)[82]. The CC appears dark in conventional OCT images owing to the flow and its inner part was selected as internal CC border (CCi).

The third choroid compartment included the choroid from the CCi line to the CSI, which was detected as a more or less sharp transition zone in the reflectivity from choroid to sclera. The hyperreflective choroidal tissue columns were also included. The sclera compartment was defined below the CSI line. The ILM, CCi, and CSI one-pixel-wide lines were drawn in a web-based, password-protected labeling tool, particularly developed for OCT B-scan image annotation (Fig. 7). B-scans were presented to the graders in a random order so that no continuous sequence of sections within an eye was possible.

The grading process of the annotated labels was conducted in three stages: (1) the tutorial set stage, (2) the test set stage, and (3) the CNN training set stage. In the tutorial set stage, each grader got instructions for the labeling task and then labeled the same set of 10 B-scans independently. In the test set stage, the data to benchmark the inter-human and human-machine variability were acquired. Table 1 describes the number of eyes and scans used for each set. All labeled B-scans were resized to 8-bit grayscale 512 × 512 pixel images.

**Ground truth generation.** Semantic segmentation maps obtained in stage three of the human labeling (900 B-scans of 36 eyes) were used to generate the ground truth for the CNN algorithm. These data were split by randomly assigning 27 eyes (675 images) to the training set and 9 eyes (225 images) to the validation set. Training and validation sets were used for training the CNN algorithm and for monitoring the learning progress during training to prevent overfitting, respectively. The training set was augmented by (1) vertically mirroring each B-scan and (2) applying a random rotation to each B-scan with a rotational angle between −8° and 8°, thereby increasing the size of the training set from 675 B-scans to 2025 B-scans.

**Ground truth size.** Initial experiments were performed to investigate the effect of ground truth size on CNN learning performance using a smaller data set to which grader 1 contributed 300 B-scans (12 eyes) and grader 2 and 3 contributed 250 B-scans (10 eyes) each. This ground truth set was randomly split into training and

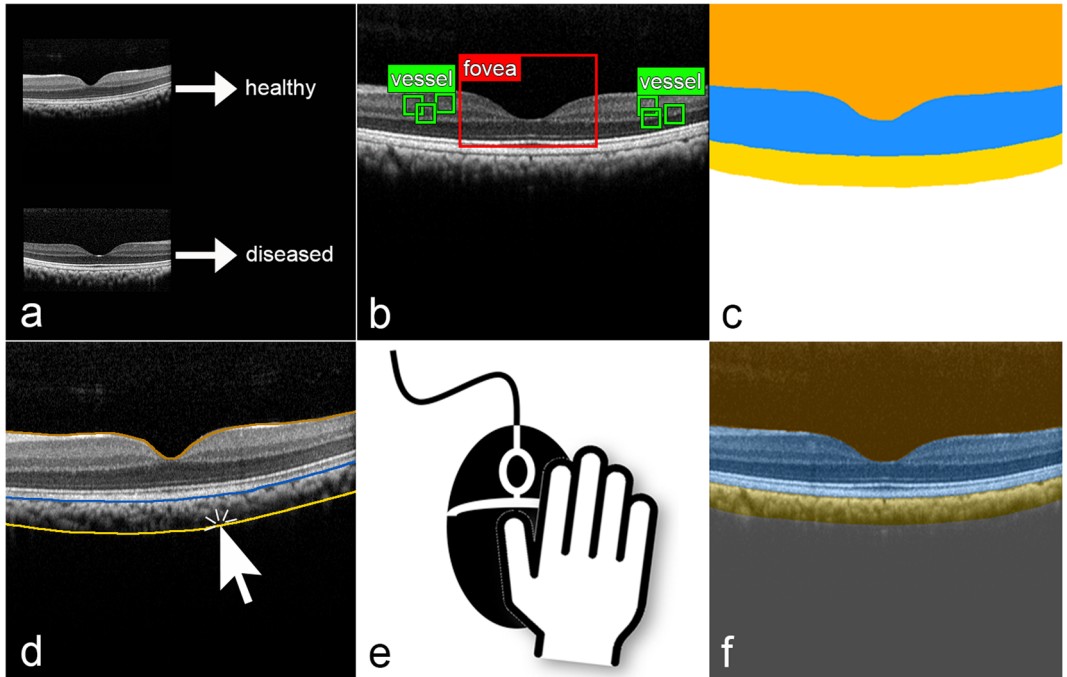

**Fig. 7 Display of difference between classification, object detection, and semantic image segmentation. a** Image classification is the task of assigning each image to one class of a set of classes (e.g. healthy or diseased). **b** Object detection is the task of identifying and localizing objects in an image (e.g. the fovea identified and located by the red rectangle or the vessels by the green rectangles). **c** Semantic image segmentation is the task of assigning each pixel of an image to one class of a set of classes. In this study each pixel of an OCT B-scan was classified as one of the four major compartments vitreous (orange), retina (blue), choroid (yellow), or sclera (white). **d** Ground truth was generated by presenting original cross-sectional B-scans to human graders in an online annotation tool. **e** Human graders drew lines between the four major compartments using a mouse-driven cursor. Pixels that lie between two lines were labeled as the compartment located between the two lines. **f** Overlay of human-generated labels with B-scan. The figure was created with Adobe Photoshop (Version 2021, licence C5004899101EDCH, Adobe, San Jose, US, and Microsoft Powerpoint 365, licence 1446383959, Microsoft, Redmond, US).

**Table 1 Overview of data sets acquired for this study.**

| Annotation stage | Amount of eyes | OCT B-scans | Grading |
|---|---|---|---|
| Tutorial set | 10 | 10 | g1, g2, g3: 10 |
| Test set | 8 | 200 | g1, g2, g3: 200 |
| CNN training and validation set | 36 | 900 | g1: 300 g2: 300 g3: 300 |

validation sets (25 and 7 eyes, respectively) and augmented applying the same strategy as for the data set of 900 B-scans described above.

**CNN architecture and training**. In this study, a U-Net[5] architecture was used with 22 convolutions, 5 transposed convolutions, and 5 skip connections, which previously proved very effective in learning semantic segmentation maps from human OCT B-scans[24].

All model parameters were initialized[83] and learned by minimizing an unweighted pixel-wise cross-entropy loss summed over the entire CNN input of $512 \times 512$ pixels. The CNN was trained with the Adam optimization algorithm on a single NVIDIA TITAN-X GPU[84,85]. A mini-batch size of eight images with an initial learning rate of $6 \times 10^{-5}$ was chosen as hyperparameters because it has proven suitable on preliminary empirical tests. Training was stopped after 1920 iterations (7.6 epochs) when the accuracy evaluated on the validation set reached a plateau.

**Comparison of human labels and CNN predictions with T-REX**. The CNN algorithm learned from ground truth generated from three independent expert graders whereby each B-scan was labeled by one human grader. The test set, however, consists of 200 B-scans of eight eyes, labeled by all of the three human graders separately.

A similar CNN has already been successfully applied and validated to humans[24], whereby the comparison was made at the level of compartments. In that previous study, the intersection over union (IOU) scores was applied. In the proposed analysis in cynomolgus monkeys, that score was changed to the Hamming distance metric to additionally compare global differences across all compartments and visualize these global differences by means of MDS plots. Hence, in this study, we quantitatively compare human labels and CNN predictions to each other: (1) across all compartments and (2) on a per-compartment level. For two semantic segmentations of a B-scan, the Hamming distance metric measures the proportion of pixels labeled differently. It thus corresponds to one minus the pixel accuracy between two sets of labels (see Fig. 2f for a visualization on a B-scan example). Significant advantages of the Hamming distance metric are that it is intuitive and allows a global quantification of differences across all labels (vitreous, retina, choroid, and sclera) at once.

Furthermore, the Hamming distance metric fulfills the criteria of a metric by mathematical definition (i.e. non-negativity, identity of indiscernibles, symmetry, and the triangle inequality). This Hamming distances metric is a natural distance function to measure the difference between two semantic segmentations across all compartments. This motivates the visualization of Hamming distances through MDS plots below. The IOU score, on the other hand, which was used in a previous study[24], does not fulfill the criteria of a metric by mathematical definition. Even though the IOU can be turned into a metric by mathematical definition by considering 1–IOU[86], also known as the Jaccard distance, the IOU and the Jaccard distance are usually calculated on a per-class level. The Hamming distance metric was chosen in the proposed analysis because (1) it is a metric by mathematical definition, (2) it is intuitively easy to understand, and (3) it takes into consideration all four classes at once.

Hamming distances were calculated between all pairs of human graders (g1 and g2, g1 and g3, g2 and g3) and between each human grader and the CNN predictions (g1 and CNN, g2 and CNN, g3 and CNN) for the 200 B-scans of the test set. See Fig. 2f for a visualization of the Hamming distance across all compartments on a single B-scan example. For each pair of gradings, the Hamming distances of all 200 B-scans of the test set were visualized using a single heatmap plot.

To calculate individual Hamming distance scores for each of the four compartments (i.e., vitreous, retina, choroid, and sclera), the semantic segmentation maps were treated as binary maps, where a pixel either does or does not belong to the respective compartment.

Mean and quartiles of Hamming distances between two gradings of the test set were calculated and visualized with boxplots. Moreover, mean Hamming distances were visualized using metric MDS[87]. MDS is a dimensionality reduction technique to visualize pairwise distances among data points by mapping those data points into a Cartesian coordinate system and preserving the original distances as well as possible. For the visualizations to be meaningful, the underlying distance must satisfy the criteria of a metric by mathematical definition (i.e. non-negativity, identity of indiscernibles, symmetry, and triangle inequality), which was fulfilled by the Hamming distance. Two-dimensional MDS plots were generated to visualize distances among the four different gradings (g1, g2, g3, CNN) of the test set labels: (1) across all compartments, (2) for the vitreous compartment, (3) for the retina compartment, (4) for the choroid compartment, and (5) for the sclera compartment.

We propose to term the approach described above as T-REX. This is, using the Hamming distance to evaluate the predictive performance of an ML model trained on ambiguous ground truth data with respect to each grader that contributed to that ground truth data set, and, subsequently, visualizing the recorded Hamming distances by heatmaps or with multidimensional scaling plots.

**Definition of neural recording**. It can be argued that CNNs currently cannot function without a computer and their internal mechanisms are often opaque. With regard to the similarities between neuroscience and ML, the observation and possible recording of the activity of such computer-based circuits could be described in a simplified way as "neural recording" (NR). Since every DL model running on a computer already would fit that description, we propose an NR framework to better understand and visualize how the recorded predictive performance of the CNN relates to the ambiguity in the ground truth data. Ideally, such a neural recording is presented in a way that is appropriate for a human cognitive scale.

**Statistics and reproducibility**. Statistical significance tests were performed to assess whether (1) inter-grader Hamming distances and (2) Hamming distances between humans and the CNN originate from distributions with different means. For this purpose, the 600 inter-grader Hamming distances (from grader pairs g1 and g2, g1 and g3, and g2 and g3) were combined into group A. The 600 Hamming distances from human–CNN comparisons (from pairs g1 and CNN, g2 and CNN, and g3 and CNN) were combined into group B. Statistical significance tests were performed to compare group A with group B (1) across all compartments, (2) for the vitreous compartment, (3) for the retina compartment, (4) for the choroid compartment, and (5) for the sclera compartment. In all five cases, the data were clearly skewed and thus unsuited for analyses with $t$-tests. The data were therefore analyzed with non-parametric, unpaired permutation tests in R[88] (URL http://www.R-project.org, last visited 10 January 2020) with the package perm[89] using 99,999 Monte Carlo simulations. The reproducibility of the ML method was previously reported[24].

To assess the reproducibility of this study's results all experiments were independently conducted on the smaller training set of 800 B-scans, which was described above. These additional experiments yielded very similar results to the results obtained on the larger set of 900 B-scans.

**Reporting summary**. Further information on research design is available in the Nature Research Reporting Summary linked to this article.

## Data availability

All source data underlying the graphs and charts presented in the main figures are available as Supplementary Data 1–5. Any remaining info can be obtained from the corresponding author upon reasonable request

## Code availability

The source code to apply the T-REX methodology described in this study is available on www.github.com/peter-maloca/T-REX and archived in Zenodo[90].

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

## Acknowledgements

We gratefully acknowledge Professor Wolfgang Drexler, Vienna, Austria, and Professor Leopold Schmetterer, Singapore Eye Research Institute, Singapore, and Institute of Molecular and Clinical Ophthalmology (IOB), Basel, Switzerland, for their valuable advice with regard to the OCT inner segmentation border of the CC. We thank for the financial support from Hoffmann-La Roche Ltd, Pharma Research and Early Development (pRED), Pharmaceutical Sciences (PS), 4070 Basel, Switzerland.

## Author contributions

A.T. and A.Y.L.: conceptualization, methodology, data curation, writing, original draft preparation, writing—review and editing, visualization; C.E.: writing—review and editing, visualization; H.P.N.S.: writing—review and editing, visualization, project administration; J.Z.-V: writing, original draft preparation, writing—review and editing, visualization; K.B.: writing—review and editing, visualization; N.D.: conceptualization, methodology, software, validation, formal analysis, investigation, resources, data curation, writing, original draft preparation, writing—review and editing, visualization, project administration, funding; P.L.M., P.M.M., P.K. and S.S.: conceptualization, methodology, software, validation, formal analysis, investigation, resources, data curation, writing, original draft preparation, writing—review and editing, visualization, project administration; P.W.H.: conceptualization, methodology, writing, original draft preparation, writing—review and editing, visualization, project administration; T.K.S.: writing—review and editing, visualization, funding, project administration; S.N.: resources, data curation, writing, original draft preparation, writing—review and editing, visualization, funding, project administration; T.S.: writing, original draft preparation, writing—review and editing.

## Competing interests

Research support was granted from Roche, Basel, Switzerland, especially with data collection and the decision to publish. Roche had no role and did not interfere in conceptualization or conduct of this study. Authors P.K. and S.S. are salaried employees of Supercomputing Systems, Zurich; Authors S.N., T.K.S., T.S. and N.D. are salaried employees of Roche, Basel, Switzerland; P.M.M. is a consultant at Roche, Basel, Switzerland. The other authors of this paper declare no competing interests. Outside of the present study, the authors declare the following competing interests: P.M.M. is a consultant at Zeiss Forum and holds intellectual properties for machine learning at MIMO AG, and VisionAI, Switzerland. A.Y.L. has received funding from Novartis, Microsoft Corporation, NVIDIA Corporation and grant number from NEI: K23EY029246. P.L.M. received funding by the German Research Foundation (grant # MU4279/2–1). C.E. and A.T. received a financial grant from the National Institute for Health Research (NIHR) Biomedical Research Centre, based at Moorfields Eye Hospital, and also from the NHS Foundation Trust and the UCL Institute of Ophthalmology. The views expressed in this article are those of the authors and not necessarily those of the National Eye Institute, NHS, the NIHR, or the Department of Health. A.T. is a consultant for Heidelberg Engineering and Optovue and has received research grant funding from Novartis and Bayer. C.E. is a consultant for Heidelberg Engineering and has received research grant funding from Novartis. J.Z.-V. is a consultant or has received travel grants of Alcon, Alimera Sciences, Allergan, Bausch&Lomb, Bayer, Brill Pharma, Novartis or Topcon.
