## [Peer Review File · Communications Biology]

Reviewers' comments:

Reviewer #1 (Remarks to the Author):

Brief Summary:

This manuscript describes an implementation of a U-Net style Convolutional Neural Network for the task of segmenting OCT images obtained from cynomolgus monkeys. The resulting segmentations are compared to ground truth annotations generated by human graders via computation of the Hamming distance, box plots and multidimensional scaling plots.

The results seem to show the CNN was capable to generalize from the training set to the test set of images, and denote a level of similarity with the grader annotations quantified by the previously described metrics and visualizations.

The manuscript also focuses on the aspect of explainability, using an analogy to a gear box and mentioning a "neural ML recording" technique.

Overall impression:

There are multiple sections of the paper that could benefit from using figures to make the employed methodology more clear and transparent, and some of the new concepts and expressions presented might be redundant with already existing nomenclature in the area of explainable AI. These issues are described in detail on the specific comments below.

Aside from other considerations, this manuscript, in its current form, notably requires extensive clarifications and/or reformulations regarding its claim to novel contributions to the field of explainable AI. This is also discussed in more detail, with relevant references, on the specific comments below.

Specific comments:

Abstract (lines 48-49): There seems to be a word missing in this sentence. Perhaps "...was surprisingly also able to [make] additional and varied predictions"?

Main (lines 58-62): The concept of segmentation in the context of Convolutional Neural Networks (CNNs) and computer vision might not be familiar to readers of all areas of knowledge. As this doesn't seem to be addressed further in the Main section of the paper, adding a short explanation regarding the meaning of this task would be interesting. An illustrative figure showing the difference between the tasks of classification, object detection and segmentation could also be included.

Main (line 73): References [26] and [27], cited here, seem to be repeated.

Main (line 75): "The ML showed..." sounds unusual. Perhaps this sentence can be complemented with terms such as algorithm/system/approach/etc. "The ML [approach] showed a performance...", for instance.

Main (line 79): relatively

Main (lines 90-99): It is unclear how the "gearbox conceptualization of ML" expands upon the already commonly used analogy of the "black-box problem" of certain ML systems. Perhaps a figure can be used to better illustrate the idea proposed by the authors.

Methods (lines 401-402): If the only optimizer used was Adam, the reference to Stochastic Gradient Descent (SGD) makes the sentence somewhat confusing. Was the classic SGD optimizer also used in addition to Adam?

Methods (line 405): What was meant by 11.1 epochs? Was training stopped midway through epoch 11? If so, please clarify and justify it. The choice of hyper-parameters (such as learning rate and mini-batch size) should also be justified in this section. Whether it was inspired by previous works or by preliminary empirical tests, this should be mentioned alongside any relevant citations.

Methods (line 424): fulfills

Methods (line 424): It is not clear what criteria the authors are referring to. It would be important to add an explanation to this paragraph, in order to clarify it.

Methods (lines 416-424): It is not entirely clear why the Hamming distance was used instead – rather than alongside – the IoU metric as in the previous study cited here. Besides this precedent cited by the authors, the IoU seems to be commonly used to assess the performance of image segmentation algorithms. As such, it could be presented in addition to the already presented Hamming distance. If there is a clear reason not to do so, it should be stated in the text.

Methods (line 432): Adding a visual example of this comparison across all compartments as opposed to individual compartments could help making this paragraph more easily understandable.

Methods (line 461): Since the “neural ML recording” method is mentioned in different points of the text, it seems important to add it to the Methods section as well, describing more clearly which parts of the methodology constitute this technique.

Results (Table 1): It might be interesting to increment the information provided by table 1 by transforming it into a heatmap plot. The numeric values could still be presented, but the color visualization makes the most variable compartments and graders more immediately apparent to the reader. Since Python seems to be one of the main languages used by the authors in this manuscript, these annotated heatmaps can be generated quickly with the Seaborn library.

Discussion (line 226): The authors mention the high reproducibility of the results. It would be interesting to justify this by including some metrics by which this reproducibility can be quantified and compared to future studies.

Discussion (lines 231-234): The sentence starting with “However” is repeated in the next paragraph.

Discussion (lines 235-239): The affirmation that the CNN is looking for a robust average among the human graders does not seem to be trivially deduced from the results. Note that most of the results presented are global analyses of the averaged behavior of the CNN over the entire test set, rather than local explanations for individual segmentations. For instance, in a clinical scenario where this model is applied, for a new unseen OCT image with no previously prepared ground truth, would it be possible to affirm whether the CNN segmentation is an average of what the opinions of graders G1 G2 and G3 would be? Whether the CNN is approximating this segmentation in particular by copying the decision making pattern of G1, G2, G3 or following its own unique combination of features? The answer to these questions doesn't seem to be trivially obtainable from the global analysis of the averaged behavior of the CNN. This qualitative affirmation could be substituted by a quantitative one discussing the generalization capabilities of the model when comparing the training, validation and test losses and performance metrics.

Discussion (line 276): Since the reasoning presented in this part is mentioned more than once on the manuscript, it is worth noting two different ideas: The training procedure of this CNN is exactly aimed at making it learn and generalize from the annotations of the three graders included in the training set. The results show that on average the CNN is doing this. This is a global behavior of the model, being discussed with a global analysis of the results over the test set. The idea, however, that the CNN is shifting gears in each image to approximate the decision making pattern of a specific grader, would be an affirmation regarding a local property, for specific individual images. It does not seem trivial to make this type of affirmation with the global analysis presented.

Discussion (line 278): While the manuscript has substantial contributions to OCT segmentation in

animals, and to the use of AI in ophthalmology in general, and these contributions stand on their own, the authors also focus on the discussing contributions to the field of explainable AI (XAI). If this is a desired focus of this manuscript, a much more in depth discussion of the XAI field should be included, to better clarify the claimed novelty of the results. A systematic review of most recent XAI methods published over the past years can be found in [1]. Some examples of attempted applications of gradient methods to semantic segmentations can be seen in [2] [3]. The idea here is to generate more local explanations, highlighting which pixels, or regions, or features of an image lead to a certain output (classification/segmentation, depending on the task). Despite it being one of the central contributions of the manuscript, the meaning of the "neural ML recordings" is not clear. The previous segmentation model for human OCT (reference [23] in the manuscript) seems to be largely similar to this one, without reference to "neural ML recording". Does this expression then refer to the MDS plots of the current manuscript? Although the MDS plots add an interesting new visualization, the similarity of the CNN predictions to the graders is also coded to a degree in the boxplots and even in the Hamming distance table (or in the case of the human OCT model, the IoU). As such it seems that possibly the paper should highlight more the contributions to OCT segmentation and the area of ophthalmology, rather than the area of XAI, unless the introduction, discussion and conclusions regarding this latter aspect are substantially reformulated to better address the topics mentioned above.

Discussion (lines 309-311): Considering the previous comments, the idea of applying the neural recording approach to rigorously test black box segmentation CNNs should be better substantiated, perhaps exemplifying how this would work in a clinical setting, with other segmentation pre-trained models.

References: Please verify if references [4] and [70] in the manuscript are repeated.

- [1] Guidotti, R., Monreale, A., Ruggieri, S., Turini, F., Giannotti, F. and Pedreschi, D. "A survey of methods for explaining black box models," ACM computing surveys, 2018.
- [2] Kristoffer, W., Kampffmeyer, M. and Jenssen, R., "Uncertainty modeling and interpretability in convolutional neural networks for polyp segmentation," 2018 IEEE 28th International Workshop on Machine Learning for Signal Processing, 2018.
- [3] Kira, V. Dibrov, A. and Myers, G. "Towards Interpretable Semantic Segmentation via Gradient-weighted Class Activation Mapping.," arXiv, 2020.

Reviewer #2 (Remarks to the Author):

The authors applied a CNN for optical coherence tomography image segmentation in animals. They calculated authors found evidence that the CNN independently leveled off in a NN approach. All in all the approaches foster the explainability aspect for the non-expert physician (which shall foster trust into such systems).

Overall the paper is an easy to read and informative contribution and is of value for the interested reader. In page 3 the authors explicitly talk about clinical routine and on page 13 they mention "explainable AI" (xAI).

Interestingly there is no pointer to the field of xAI - which the authors should do e.g. on page 3, here is a highly cited paper general introductory paper into this field [1]:

- [1] Holzinger, A., Kieseberg, P., Weippl, E. & Tjoa, A. M. 2018. Current Advances, Trends and Challenges of Machine Learning and Knowledge Extraction: From Machine Learning to Explainable AI. Springer Lecture Notes in Computer Science LNCS 11015. Cham: Springer, pp. 1-8, doi:10.1007/978-3-319-99740-7-1

Even more important is the following issue:

Whilst explainable AI (XAI) deals with the implementation of transparency and traceability of statistical black-box machine learning methods, particularly deep learning approaches - as outlined in this paper - there is in certain domains (such as the medical domain, and particularly the clinical domain which the authors mentioned explicitly on page 3 which the authors mentioned explicitly, and again on page 13), an pressing need to go beyond explainable AI; For example, to reach a level of explainable medicine (!) there is a crucial need for causability. Causability [2] is different from Causality (in the sense of Judea Pearl) but closely connected. In the same way that usability encompasses measurements for the quality of use, causability encompasses measurements for the quality of explanations produced by explainable AI methods (e.g. a heatmap – see below). Causability is the property of a human (=human intelligence), whereas explainability is a property of a system (=artificial intelligence).

In the medical domain and particularly in the clinical domain it is of supreme importance to enable the domain expert to understand, why (!) an algorithm came up with a certain result (this is necessary e.g. due to raising legal issues). With certain explainable-AI methods, such as layer-wise relevance propagation [3], relevant parts of inputs to, and representations in, a neural network which caused a result, can be highlighted (with a heatmap). However, this is only a first – important – but only first step to ensure that end users, e.g., medical professionals (the human!), assume responsibility for decision making with AI. The backbone for this approach is interactive ML [4], which adds the component of human expertise to AI/ML processes by enabling them to re-enact and retrace the results on demand, e.g. let them check it for plausibility.

[2] Holzinger, A., Langs, G., Denk, H., Zatloukal, K. & Mueller, H. 2019. Causability and Explainability of Artificial Intelligence in Medicine. Wiley Interdisciplinary Reviews: Data Mining and Knowledge Discovery, 9, (4), doi:10.1002/widm.1312.

[3] Seegerer, P., Binder, A., Saitenmacher, R., Bockmayr, M., Alber, M., Jurmeister, P., Klauschen, F. & Müller, K.-R. 2020. Interpretable Deep Neural Network to Predict Estrogen Receptor Status from Haematoxylin-Eosin Images. In: Holzinger, Andreas, Goebel, Randy, Mengel, Michael & Müller, Heimo (eds.) Artificial Intelligence and Machine Learning for Digital Pathology: State-of-the-Art and Future Challenges. Cham: Springer International Publishing, pp. 16-37, doi:10.1007/978-3-030-50402-1_2.

[4] Holzinger, A. 2016. Interactive Machine Learning for Health Informatics: When do we need the human-in-the-loop? Brain Informatics, 3, (2), 119-131, doi:10.1007/s40708-016-0042-6.

Minor comments:

The references are inconsistent, some are with full name [73]

Reference [50] is incomplete

please check carefully ALL references

General:

The article is interesting and relevant and this reviewer would recommend acceptance given the additional issues to be addressed as outlined above.

Reviewer #3 (Remarks to the Author):

In this manuscript, the authors developed a machine learning (ML) platform for retina OCT image segmentation and proposed a “gearbox-conceptualization” of ML to uncover the ML learning strategy in order to provide knowledge of its workflows. The results show that the differences between the platform and human graders were smaller than the total inter-grader variability.

Specific comments.

- 1.Originality: The methodology presented in this work is not novel and based on the existing advances in machine learning.
- 2.The authors trained a deep CNN model which consisted of 22 convolutions, 5 transposed convolutions and 5 skip connections to segment OCT images automatically. However, the authors used only 8 and 32 eyes for testing and the ground truth generation for CNN algorithm. The OCT scan for each eye was exported to a stack of 25 B-Scans. The dataset for this kind of learning-based method seems rather small, which makes it difficult to interpret the validation of results.
- 3.The CNN algorithm learned from ground truth generated from three independent expert graders whereby each B-Scan was labeled by one human grader. The results of CNN algorithm may be greatly affected by the human grader who labeled more B-scans. The authors need to add more labeling details by experts.
- 4.In the "neural recording" method section, the authors made explanatory analysis of the model through the balanced performance for CNN results, without explaining the internal working mechanism of the model, which is not consistent with the "ML- gearbox" described in the manuscript.

Minor comments:

- 1.How do the authors propose to implement the AI system for real-world application.
- 2.The statistics used should be defined in the paper. The 95% confidence interval should be reported.
- 3.Illustration of the OCT segmentations was missed in Figure 1.
- 4.The manuscript needs to be polished.

Basel, 24 October 2020

Dear reviewers

Thank you very much for the valuable time you have invested in improving our manuscript. We agree with all comments and have made every effort to implement your suggestions for improvement! The quality of the manuscript has increased considerably with your help and we have learned a lot from you on behalf of the whole team I thank you!

Peter Maloca

Shortend title:

“Unraveling the deep learning gearbox in optical coherence tomography image segmentation towards explainable artificial intelligence”

Shared comments by all reviewers

Comments to reviewer #1

Brief Summary: This manuscript describes an implementation of a U-Net style Convolutional Neural Network for the task of segmenting OCT images obtained from cynomolgus monkeys. The resulting segmentations are compared to ground truth annotations generated by human graders via computation of the Hamming distance, box plots and multidimensional scaling plots.

The results seem to show the CNN was capable to generalize from the training set to the test set of images, and denote a level of similarity with the grader annotations quantified by the previously described metrics and visualizations.

The manuscript also focuses on the aspect of explainability, using an analogy to a gear box and mentioning a “neural ML recording” technique.

Overall impression:

There are multiple sections of the paper that could benefit from using figures to make the employed methodology more clear and transparent, and some of the new concepts and expressions presented might be redundant with already existing nomenclature in the area of explainable AI. These issues are described in detail on the specific comments below.

Aside from other considerations, this manuscript, in its current form, notably requires extensive clarifications and/or reformulations regarding its claim to novel contributions to the field of explainable AI. This is also discussed in more detail, with relevant references, on the specific comments below.

Comment 1: Abstract (lines 48-49): There seems to be a word missing in this sentence. Perhaps “...was surprisingly also able to [make] additional and varied predictions”?

Response 1: Thank you for pointing the missing word out. We have adjusted to “..was surprisingly also able to make additional and varied predictions..”

Comment 2: Main (lines 58-62): The concept of segmentation in the context of Convolutional Neural Networks (CNNs) and computer vision might not be familiar to readers of all areas of knowledge. As this doesn't seem to be addressed further in the Main section of the paper, adding a short explanation regarding the meaning of this task would be interesting. An illustrative figure showing the difference between the tasks of classification, object detection and segmentation could also be included.

Response 2: That is a very good comment. We have now included an additional Fig. 6 which illustrates the different tasks for a better understanding:

Figure 6. Difference between classification, object detection, and semantic image segmentation. **a**, image classification is the task of assigning each image to one class of a set of classes (e.g. healthy or diseased). **b**, object detection is the task of identifying and localizing objects in an image (e.g. the fovea identified and located by the red rectangle or the vessels by the green rectangles). **c**, semantic image segmentation is the task of assigning each pixel of an image to one class of a set of classes. In this study each pixel of an OCT B-scan was classified as one of the four major compartments vitreous (orange), retina (blue), choroid (yellow), or sclera (white). **d**, ground truth was generated by presenting original cross-sectional B-scans to human graders in an online annotation tool. **e**, human graders drew lines between the four major compartments using a mouse-driven cursor. Pixels that lie between two lines were labeled as the compartment located between the two lines. **f**, overlay of the human-generated labels with B-scan.

Comment 3: Main (line 73): References [26] and [27], cited here, seem to be repeated.

Response 3: all references have been adjusted

Comment 4: Main (line 75): “The ML showed...” sounds unusual. Perhaps this sentence can be complemented with terms such as algorithm/system/approach/etc. “The ML [approach] showed a performance...”, for instance.

Response 4: Thank you for pointing this out. We have completed the wording to “The ML approach showed a performance in making a referral recommendation that was at least as good as human experts”

Comment 5: Main (line 79): relatively

Response 5: We have adjusted to “take place in a relatively narrow”

Comment 6: Main (lines 90-99): It is unclear how the “gearbox conceptualization of ML” expands upon the already commonly used analogy of the “black-box problem” of certain ML systems. Perhaps a figure can be used to better illustrate the idea proposed by the authors.

Response 6: The gearbox has been explained in more details. Please see below the Response #4 of reviewer 3.

Comment 7: Methods (lines 401-402): If the only optimizer used was Adam, the reference to Stochastic Gradient Descent (SGD) makes the sentence somewhat confusing. Was the classic SGD optimizer also used in addition to Adam?

Response 7: It’s true that this sentence is confusing. We reworded to “The CNN was trained with the Adam optimization algorithm^{72,73} using a mini-batch size of eight images with an initial learning rate of 6×10^{-5} on a single NVIDIA TITAN-X GPU.”

Comment 8: Methods (line 405): What was meant by 11.1 epochs? Was training stopped midway through epoch 11? If so, please clarify and justify it.

The choice of hyper-parameters (such as learning rate and mini-batch size) should also be justified in this section. Whether it was inspired by previous works or by preliminary empirical tests, this should be mentioned alongside any relevant citations.

Response 8: We agree with this comment and improved the description of the CNN learning process in this section accordingly to:

“All model parameters were initialized⁷¹ and learned by minimizing an unweighted pixel-wise cross-entropy loss summed over the entire CNN input of 512 pixels \times 512 pixels. The CNN was trained with the Adam optimization algorithm^{72,73} on a single NVIDIA TITAN-X GPU. A mini-batch size of eight images with an initial learning rate of 6×10^{-5} was chosen as hyperparameters because it has proven suitable on preliminary empirical tests. Training was stopped after 1920 iterations (7.6 epochs) when the accuracy evaluated on the validation set reached a plateau.”

We stopped training after the validation-set accuracy has reached a plateau, which was the case after 7.6 epochs. We could possibly also have stopped training after an exact number of n (7 or 8) epochs but in preliminary results it didn’t matter when we stopped training (halfway through an epoch or after n epochs). Note that the number of epochs changed from

11.1 to 7.6 since in the current version of the manuscript we retrained the model with further data as suggested by reviewer #3.

Comment 9: Methods (line 424): fulfils

Response 9: We adjusted the text accordingly.

Comment 10: Methods (line 424): It is not clear what criteria the authors are referring to. It would be important to add an explanation to this paragraph, in order to clarify it.

Response 10: We agree with the point highlighted by the reviewer. Our wording used in the first version of the manuscript was not precise enough to express our intended line of reasoning. In this paragraph, we are referring to the criteria of a metric by mathematical definition (non-negativity, identity of indiscernibles, symmetry, and the triangle inequality) considering all four eye compartments at once. We improved the wording accordingly and added an additional paragraph as explanation.

Fulfilling the criteria of a metric by mathematical definition makes the Hamming distance a natural distance function to measure the difference between two semantic segmentations. We think this justifies the visualization of the Hamming distances through multidimensional scaling plots later in the paper. This is because we think that the mathematical definition of a metric represents what we intuitively consider “natural” about a “distance”. The frequently used IoU metric, on the other hand, is not a metric by mathematical definition. However, $1 - \text{IoU}$, also known as Jaccard distance, is a metric by mathematical definition [1001]. But IoU and the Jaccard distance are usually calculated on a per-class level. Our reasoning to use the Hamming distance in this work here is that we want a distance function that (1) is a metric by mathematical definition, (2) is intuitively easy to understand, and (3) takes into consideration all four classes at once.

[1001] S. Kosub. A note on the triangle inequality for the jaccard distance. arXiv:1612.02696, 2016.

Comment 11: Methods (lines 416-424): It is not entirely clear why the Hamming distance was used instead – rather than alongside – the IoU metric as in the previous study cited here. Besides this precedent cited by the authors, the IoU seems to be commonly used to assess the performance of image segmentation algorithms. As such, it could be presented in addition to the already presented Hamming distance.

If there is a clear reason not to do so, it should be stated in the text.

Response 11: Thank you for this comment. We acknowledge that we didn't explain carefully enough why we chose the Hamming distance over the IoU score. We understand that this is critical since most results, which we present and discuss, are based on the Hamming distance. We think that our reply to Comment 10 covers some of the concerns raised in reviewer's Comment 11 already.

After carefully reviewing the appreciated suggestions of the reviewer, we decided not to include the IoU (or the Jaccard distance) alongside the Hamming distance. The IoU since it's not a metric by mathematical definition. A visualization of the IoU through multidimensional scaling would therefore not be meaningful. We believe that the visualization of the differences between semantic segmentation by means of multidimensional scaling plots is one of the significant contributions of this paper. Since this is not possible for the IoU, we decided not to include it. Regarding the Jaccard distance, while including the Jaccard

distance on a per-class level might give the reader useful additional information and could also be visualized through multidimensional scaling plots, we think that including a “global” Jaccard distance would be more difficult to motivate and understand. To calculate a global Jaccard distance, we would need to calculate some form of average, e.g. the mean over the four per-class Jaccard distances. To our knowledge, it is not common to calculate and discuss such a coefficient in the context of semantic image segmentations. Besides, we think that such a coefficient would not be particularly intuitive to understand. For these reasons we decided against the inclusion of the IoU and Jaccard distance. We describe the reasoning behind this decision also in the additional paragraph, which we have added to the paper’s method section.

Comment 12: Methods (line 432): Adding a visual example of this comparison across all compartments as opposed to individual compartments could help making this paragraph more easily understandable.

Response 12: We liked the idea of adding a visual example to highlight this comparison visually. Therefore, we added figure 1f. We also reworded this paragraph since it was not straightforward to understand. It reads now:

“Hamming distances were calculated between all pairs of human graders (g1 and g2, g1 and g3, g2 and g3) and between each human grader and the CNN predictions (g1 and CNN, g2 and CNN, g3 and CNN) for the 200 B-scans of the test set: (1) across all compartments, (2) for the vitreous compartment, (3) for the retina compartment, (4) for the choroid compartment, and (5) for the sclera compartment. Hamming distances were calculated on a per B-scan basis (see Figure 1 (f) for a visualization across all compartments on a single B-scan example). Hamming distances across all compartments for each B-scan of the test set were visualized using a heatmap plot.”

Comment 13: Methods (line 461): Since the “neural ML recording” method is mentioned in different points of the text, it seems important to add it to the Methods section as well, describing more clearly which parts of the methodology constitute this technique.

Response 13: Thank you for pointing out that the term “neural ML recording” is not well understood and not a standard term in machine learning, yet. The term was chosen based on overlap between neuroscience and ML. We would like to introduce the idea of revealing insights (“recordings”) into the neural network training in the sense of explainable AI (XAI). We therefore included a section in the introduction that reviews XAI literature. Moreover, we explain our proposed XAI method in more detail and state how it relates to previous work. Consequently, we adapted our title to:

“Unraveling the deep learning gearbox in optical coherence tomography image segmentation towards explainable artificial intelligence”.

We have now integrated a definition of “neural recording” in the domain of machine learning in the methods and explained its application in the methods and discussion in much more details:

“Definition of neural recording

With regard to the overlap between neuroscience and ML, it can be argued that CNNs currently cannot function without a computer and their internal mechanisms are often

opaque. The observation and possible recording of the activity of such computer-based circuits could be described in a simplified way as "neural recording". Since every deep learning model running on a computer already would fit that description, we propose a neural recording framework to better understand and visualize how the recorded predictive performance of the CNN relates to the ambiguity in the ground truth data. Ideally, such a neural recording is presented in a way that is appropriate for a human cognitive scale."

Comment 14: Results (Table 1): It might be interesting to increment the information provided by table 1 by transforming it into a heatmap plot. The numeric values could still be presented, but the color visualization makes the most variable compartments and graders more immediately apparent to the reader.

Since Python seems to be one of the main languages used by the authors in this manuscript, these annotated heatmaps can be generated quickly with the Seaborn library.

Response 14: Thank you for this suggestion. We liked this idea and thus we generated the table as a heatmap with the Python library Seaborn. Indeed, this highly increases the information content conveyed by Table 1.

Comment 15: Discussion (line 226): The authors mention the high reproducibility of the results. It would be interesting to justify this by including some metrics by which this reproducibility can be quantified and compared to future studies.

Response 15: Actually, we meant reproducibility with respect to the previously published validation study [1003], which had a similar focus but was done on human data instead of monkey data. Probably, our sentence was unclear. Hence, we reformulated to

"The translation of a previously developed ML framework in humans²³ to animals was successful."

Regarding the inclusion of a metric for comparison among other studies: we compare our study to [1003] by discussing mean Hamming distances over all compartments further bellow in the discussion.

[1003] Maloca, P.M., et al. Validation of automated artificial intelligence segmentation of optical coherence tomography images. PLOS ONE (2019), <https://doi.org/10.1371/journal.pone.0220063>

Comment 16: Discussion (lines 231-234): The sentence starting with "However" is repeated in the next paragraph.

Response 16: We have deleted this sentence for a better readability "Furthermore, our results showed for the first time that the CNN is actively looking for a sort of robust average among all the human graders."

Comment 17: Discussion (lines 235-239): The affirmation that the CNN is looking for a robust average among the human graders does not seem to be trivially deduced from the results. Note that most of the results presented are global analyses of the averaged behavior of the CNN over the entire test set, rather than local explanations for individual segmentations. For instance, in a clinical scenario where this model is applied, for a new unseen OCT image with no previously prepared ground truth, would it be possible to affirm whether the CNN segmentation is an average of what the opinions of graders G1 G2 and G3 would be? Whether the CNN is approximating this segmentation in particular by copying the

decision making pattern of G1, G2, G3 or following its own unique combination of features? The answer to these questions doesn't seem to be trivially obtainable from the global analysis of the averaged behavior of the CNN.

This qualitative affirmation could be substituted by a quantitative one discussing the generalization capabilities of the model when comparing the training, validation and test losses and performance metrics.

Response 17: We agree with the reviewer's comment and appreciate it. It's true that we can't trivially deduce that the CNN is looking for a robust average among the three human graders based on the global analysis alone. Thus, we included an analysis based on a per B-scan basis and we added an additional heatmap plot (see below). The figure visualizes the recorded Hamming distance for each of the 200 B-scans of the test set for each of the six comparisons (g1,g2;g1,g3;g2,g3;g1,cnn;g2,cnn;g3,cnn). We believe that this figure shows that the CNN is looking for an average among the three human graders for most of the 200 B-scans of the test set (indicated by the predominantly green color in the bottom three rows of the figure).

Comment 18: Discussion (line 276): Since the reasoning presented in this part is mentioned more than once on the manuscript, it is worth noting two different ideas:

The training procedure of this CNN is exactly aimed at making it learn and generalize from the annotations of the three graders included in the training set. The results show that on average the CNN is doing this. This is a global behavior of the model, being discussed with a global analysis of the results over the test set.

The idea, however, that the CNN is shifting gears in each image to approximate the decision making pattern of a specific grader, would be an affirmation regarding a local property, for specific individual images. It does not seem trivial to make this type of affirmation with the global analysis presented.

Response 18: The concerns raised by the reviewer in this comment are justified and valid. Indeed, these are two different ideas. We addressed the issue of discussing the CNN's generalization abilities in the context of a global analysis only in our response to comment 17 already. In particular, we added an analysis that is based on a per B-scan basis. Regarding the gear shifting analogy, we reformulated to:

"Dependent on the compartment, i.e., data label, the CNN judges the importance of the labels of the three human graders during training differently. For the vitreous and retina

compartments, g2 and g3 produced labels relatively similarly, and g1 produced labels relatively differently. During training the CNN seems to pay more attention to the labels of the two graders who labeled similarly since the mean Hamming distance across the test set of 200 B-scans is closer to g2 and g3 than to g1 (Table 1). On the other hand, for the choroid and the sclera compartments, g1 and g2 labeled relatively similarly and g2 and g3 labeled relatively similarly. But the labels of g1 and g3 were relatively different. In this case, the CNN learned to make predictions that are closer to g1 than to g2 and g3. This behavior can be compared to a gear shift. Depending on the compartment, i.e. the data label, the CNN applies a different learning strategy with respect to the ground truth data. This demonstrates the importance of employing multiple independent graders for CNN training."

Comment 19: Discussion (line 278): While the manuscript has substantial contributions to OCT segmentation in animals, and to the use of AI in ophthalmology in general, and these contributions stand on their own, the authors also focus on the discussing contributions to the field of explainable AI (XAI).

If this is a desired focus of this manuscript, a much more in depth discussion of the XAI field should be included, to better clarify the claimed novelty of the results. A systematic review of most recent XAI methods published over the past years can be found in [1]. Some examples of attempted applications of gradient methods to semantic segmentations can be seen in [2] [3]. The idea here is to generate more local explanations, highlighting which pixels, or regions, or features of an image lead to a certain output (classification/segmentation, depending on the task).

Despite it being one of the central contributions of the manuscript, the meaning of the "neural ML recordings" is not clear. The previous segmentation model for human OCT (reference [23] in the manuscript) seems to be largely similar to this one, without reference to "neural ML recording". Does this expression then refer to the MDS plots of the current manuscript? Although the MDS plots add an interesting new visualization, the similarity of the CNN predictions to the graders is also coded to a degree in the boxplots and even in the Hamming distance table (or in the case of the human OCT model, the IoU).

As such it seems that possibly the paper should highlight more the contributions to OCT segmentation and the area of ophthalmology, rather than the area of XAI, unless the introduction, discussion and conclusions regarding this latter aspect are substantially reformulated to better address the topics mentioned above.

- [1] Guidotti, R., Monreale, A., Ruggieri, S., Turini, F., Giannotti, F. and Pedreschi, D. "A survey of methods for explaining black box models," ACM computing surveys, 2018.
- [2] Kristoffer, W., Kampffmeyer, M. and Jenssen, R., "Uncertainty modeling and interpretability in convolutional neural networks for polyp segmentation," 2018 IEEE 28th International Workshop on Machine Learning for Signal Processing, 2018.
- [3] Kira, V. Dibrov, A. and Myers., G. "Towards Interpretable Semantic Segmentation via Gradient-weighted Class Activation Mapping.," arXiv, 2020.

Response 19: Thank you for the insightful discussion on the embedding of our proposed work. The contributions are indeed twofold: 1) the training of an OCT segmentation model applied to monkeys for the first time; 2) the proposing of a new XAI method (X-REG) that visualizes the network variations learned from ambiguous ground truth data. To make this clearer, we embedded our manuscript further in the context of XAI. Therefore, we relate our work to the XAI references below including suggested references by the reviewers. Typical methods in XAI mainly focus on post-hoc prediction visualization of how a network makes predictions by highlighting regions in the input that are relevant. In our proposed approach,

we trained a network with ground truth of different human graders and then visualized the trained model variability between human graders with multi-dimensional scaling plots using the Hamming distance. Hence, we do not highlight relevant regions of the input images, but similarities of the network to the varying human graders the network has learned from.

The related work embedding reads now as follows:

“Investigating the AI black box has become known as explainable AI (XAI), which provides tools that reveal the AI decisions. The call for transparency of AI models is especially high in medicine, where uncertainty, ambiguity and unknown are inherent to the discipline. XAI distinguishes taxonomies such as: understanding referring to comprehend the inner mechanisms of an AI model; explaining revealing “why” a machine technically decided for an outcome based on a collection of features that contributed to the AI decision; and interpreting mapping the abstract (and technical) XAI concepts to a human understandable format. While humans often cannot explain the reasoning behind a decision, understanding an AI model’s decision process will provide confidence, trust, and acceptance of the machine. Further trust is achieved by causability approaches, which measure the quality of explanations produced by XAI techniques in the human intelligence domain, e.g., with the System Causability Scale. In this work, we evaluate explainability with a new post-hoc XAI technique, which we suggest terming Explainable Relevance Evaluation of Graders (X-REG). Post-hoc XAI techniques such as LIME, BETA, GradCAM, LRP, Deep Taylor decomposition or TGAV highlight and visualize regions of the input data that lead to relevant prediction decisions after the neural network training process (post-hoc). Whereas in the proposed work, we trained a CNN from ambiguous ground truth consisting of labels from three human graders that hence acted as three CNN “teachers”. Using our novel XAI technique X-REG, we propose to post-hoc evaluate the variability of the CNN predictions relative to the human graders by using Hamming distances and then visualizing the measured variability with multi-dimensional scaling plots. Using this approach, we highlight similarities of the trained CNN to the varying human graders (or “teachers”) that the CNN has learned from. In particular, we applied the proposed XAI method X-REG to OCT image segmentation. We evaluated and visualized for the first time how a CNN trained from ambiguous ground truth data independently positioned itself between the human graders with respect to how the humans labeled the ground truth.”

The discussion embedding reads now as follows:

“Above all and despite all limits, in medicine, a doctor will only use such an AI system for diagnosis if he can understand and comprehend the internal AI processing. More importantly, a physician will only make a clinical decision based on a recommendation of such an AI system if she can fully identify with the AI. A subset of XAI methods aims at revealing post-hoc insights into “why” a machine has taken a certain decision. While well-known post-hoc approaches such as LRP or GradCAM visualize relevant regions in the input data, X-REG, our proposed XAI method, evaluated and visualized similarities between the CNN predictions and the gradings of multiple humans that the CNN has learned from. Therefore, this study contributes to a better explainability in the application of AI, such that a resulting DL model can be better appreciated: X-REG, our proposed, novel XAI technique, was successfully applied to OCT image segmentation and effectively visualized the machine decision taking from ambiguous ground truth data. X-REG can provide a rigorous evaluation and re-calibration tool to incorporate new DL standards. In a more general sense, it can increase the quality of explanations that are based on DL systems, which increases causability^{ref}. This in

turn can promote safety for doctors and patients. Accordingly, the proposed post-hoc XAI approach X-REG is expected to enable data scientists to model more transparent DL systems. In return, this leads to more trust in trained DL models by physicians, which utilize DL for data-supported clinical decisions.”

Castelvecchi, Can we open the black box of AI? Nature News, 538 (7623) (2016), p. 20

Guidotti, R., Monreale, A., Ruggieri, S., Turini, F., Giannotti, F. and Pedreschi, D. "A survey of methods for explaining black box models," ACM computing surveys, 2018.

Z.C. Lipton, The mythos of model interpretability, Queue, 16 (3) (2018), pp. 30:31-30:57

D. Gunning, Explainable artificial intelligence (xAI), Technical Report, Defense Advanced Research Projects Agency (DARPA) (2017)

Holzinger, A., Kieseberg, P., Weippl, E. & Tjoa, A. M. 2018. Current Advances, Trends and Challenges of Machine Learning and Knowledge Extraction: From Machine Learning to Explainable AI. Springer Lecture Notes in Computer Science LNCS 11015. Cham: Springer, pp. 1-8, doi:10.1007/978-3-319-99740-7-1

Alejandro Barredo Arrieta, Natalia Díaz-Rodríguez, Javier Del Ser, Adrien Bennetot, Siham Tabik, Alberto Barbado, Salvador Garcia, Sergio Gil-Lopez, Daniel Molina, Richard Benjamins, Raja Chatila, Francisco Herrera, Explainable Artificial Intelligence (XAI): Concepts, taxonomies, opportunities and challenges toward responsible AI, Information Fusion, Volume 58, 2020, Pages 82-115, <https://doi.org/10.1016/j.inffus.2019.12.012>

E. Tjoa, C. Guan, A survey on explainable artificial intelligence (XAI): Towards medical XAI, 2019.

Robert Müller, Methods for interpreting and understanding deep neural networks, Digital Signal Processing, Volume 73, 2018, Pages 1-15, <https://doi.org/10.1016/j.dsp.2017.10.011>.

Holzinger, A., Langs, G., Denk, H., Zatloukal, K. & Mueller, H. 2019. Causability and Explainability of Artificial Intelligence in Medicine. Wiley Interdisciplinary Reviews: Data Mining and Knowledge Discovery, 9, (4), doi:10.1002/widm.1312.

Holzinger, A., Carrington, A. & Müller, H. Measuring the Quality of Explanations: The System Causability Scale (SCS). Künstl Intell 34, 193–198 (2020). <https://doi.org/10.1007/s13218-020-00636-z>

M.T. Ribeiro, S. Singh, C. Guestrin, Why should I trust you?: Explaining the predictions of any classifier, ACM SIGKDD International Conference on Knowledge Discovery and Data Mining, ACM (2016), pp. 1135-1144

Lakkaraju, H., Kamar, E., Caruana, R., & Leskovec, J. (2017). Interpretable and explorable approximations of black box models. arXiv:1707.01154.

R.R. Selvaraju, M. Cogswell, A. Das, R. Vedantam, D. Parikh, D. Batra, Grad-cam: Visual explanations from deep networks via gradient-based localization, Proceedings of the IEEE International Conference on Computer Vision (2017), pp. 618-626

Kristoffer, W., Kampffmeyer, M. and Jenssen, R., "Uncertainty modeling and interpretability in convolutional neural networks for polyp segmentation," 2018 IEEE 28th International Workshop on Machine Learning for Signal Processing, 2018.

Kira, V. Dibrov, A. and Myers, G. "Towards Interpretable Semantic Segmentation via Gradient-weighted Class Activation Mapping.," arXiv, 2020.

On Pixel-Wise Explanations for Non-Linear Classifier Decisions by Layer-Wise Relevance Propagation, Bach S, Binder A, Montavon G, Klauschen F, Müller KR, et al. (2015) On Pixel-Wise Explanations for Non-Linear Classifier Decisions by Layer-Wise Relevance Propagation. PLOS ONE 10(7): e0130140. <https://doi.org/10.1371/journal.pone.0130140>

Seegerer, P., Binder, A., Saitenmacher, R., Bockmayr, M., Alber, M., Jurmeister, P., Klauschen, F. & Müller, K.-R. 2020. Interpretable Deep Neural Network to Predict Estrogen Receptor Status from Haematoxylin-Eosin Images. In: Holzinger, Andreas, Goebel, Randy, Mengel, Michael & Müller, Heimo (eds.) Artificial Intelligence and Machine Learning for Digital Pathology: State-of-the-Art and Future Challenges. Cham: Springer International Publishing, pp. 16-37, doi:10.1007/978-3-030-50402-1_2.

G. Montavon, S. Lapuschkin, A. Binder, W. Samek, K.-R. Müller, Explaining nonlinear classification decisions with deep Taylor decomposition, Pattern Recognition, 65 (2017), pp. 211-222

B. Kim, M. Wattenberg, J. Gilmer, C. Cai, J. Wexler, F. Viegas, R. Sayres, Interpretability beyond feature attribution: Quantitative testing with concept activation vectors (TCAV), 2017.

Comment 20: Discussion (lines 309-311): Considering the previous comments, the idea of applying the neural recording approach to rigorously test black box segmentation CNNs should be better substantiated, perhaps exemplifying how this would work in a clinical setting, with other segmentation pre-trained models.

Response 20: As explained in the earlier comments, we embedded our work more clearly into existing XAI research.

Comment 21: References: Please verify if references [4] and [70] in the manuscript are repeated.

Response 21: We are sorry that our reference managing software (EndNote) did not correct all references. All references have now been adjusted.

Comments to Reviewer #2

The authors applied a CNN for optical coherence tomography image segmentation in animals. They calculated authors found evidence that the CNN independently leveled off in a NN approach. All in all the approaches foster the explainability aspect for the non-expert physician (which shall foster trust into such systems).

Overall the paper is an easy to read and informative contribution and is of value for the interested reader. In page 3 the authors explicitly talk about clinical routine and on page 13 they mention "explainable AI" (xAI).

Comment 1: Interestingly there is no pointer to the field of xAI - which the authors should do e.g. on page 3, here is a highly cited paper general introductory paper into this field [1]:
[1] Holzinger, A., Kieseberg, P., Weippl, E. & Tjoa, A. M. 2018. Current Advances, Trends and Challenges of Machine Learning and Knowledge Extraction: From Machine Learning to Explainable AI. Springer Lecture Notes in Computer Science LNCS 11015. Cham: Springer, pp. 1-8, doi:10.1007/978-3-319-99740-7-1
Even more important is the following issue:

Whilst explainable AI (XAI) deals with the implementation of transparency and traceability of statistical black-box machine learning methods, particularly deep learning approaches - as outlined in this paper - there is in certain domains (such as the medical domain, and particularly the clinical domain which the authors mentioned explicitly on page 3 which the authors mentioned explicitly, and again on page 13), an pressing need to go beyond explainable AI; For example, to reach a level of explainable medicine (!) there is a crucial need for causability. Causability [2] is different from Causality (in the sense of Judea Pearl) but closely connected. In the same way that **usability** encompasses measurements for the quality of use, **causability** encompasses measurements for the quality of explanations produced by explainable AI methods (e.g. a heatmap – see below). Causability is the property of a human (=human intelligence), whereas explainability is a property of a system (=artificial intelligence).

In the medical domain and particularly in the clinical domain it is of supreme importance to enable the domain expert to understand, why (!) an algorithm came up with a certain result (this is necessary e.g. due to raising legal issues). With certain explainable-AI methods, such as layer-wise relevance propagation [3], relevant parts of inputs to, and representations in, a neural network which caused a result, can be highlighted (with a heatmap). However, this is only a first – important – but only first step to ensure that end users, e.g., medical professionals (the human!), assume responsibility for decision making with AI. The backbone for this approach is interactive ML [4], which adds the component of human expertise to AI/ML processes by enabling them to re-enact and retrace the results on demand, e.g. let them check it for plausibility.

[2] Holzinger, A., Langs, G., Denk, H., Zatloukal, K. & Mueller, H. 2019. Causability and Explainability of Artificial Intelligence in Medicine. Wiley Interdisciplinary Reviews: Data Mining and Knowledge Discovery, 9, (4), doi:10.1002/widm.1312.

[3] Seegerer, P., Binder, A., Saitenmacher, R., Bockmayr, M., Alber, M., Jurmeister, P., Klauschen, F. & Müller, K.-R. 2020. Interpretable Deep Neural Network to Predict Estrogen Receptor Status from Haematoxylin-Eosin Images. In: Holzinger, Andreas, Goebel, Randy, Mengel, Michael & Müller, Heimo (eds.) Artificial Intelligence and Machine Learning for Digital Pathology: State-of-the-Art and Future Challenges. Cham: Springer International Publishing, pp. 16-37, doi:10.1007/978-3-030-50402-1_2.

[4] Holzinger, A. 2016. Interactive Machine Learning for Health Informatics: When do we need the human-in-the-loop? Brain Informatics, 3, (2), 119-131, doi:10.1007/s40708-016-0042-6.

Response 1: Thank you for the valuable pointers to existing XAI researchers. These references are very instructive. Therefore, we are thankful and we added a literature review to recent XAI approaches and explained the varying aspects of XAI that are researched – including the above references. Our proposed XAI approach visualizes post-hoc “why” a network took a certain decision based on ambiguous ground truth data. In the introduction and discussion, we illustrated more clearly how our work relates to and differs from others. In contrast to previous XAI approaches that visualize decision-relevant regions in input data, we propose a method, which we propose to term X-REG, to highlight variations in network predictions that were learned from ambiguous ground truth data post-hoc. The research pointed out in causality is highly relevant but goes even one step further than what we proposed. While we also use heatmaps, we did not yet analyze the usability of the quality (causability) of our XAI approach. However, in the discussion we mention the potential of this new method to be useful in the context of causability. It is definitely a relevant future aspect of research to be looked even further into.

Comment 2: The references are inconsistent, some are with full name [73]

Comment 3: Reference [50] is incomplete

Comment 4: please check carefully ALL references

Comment 5: General: The article is interesting and relevant and this reviewer would recommend acceptance given the additional issues to be addressed as outlined above.

Response 2-5. The references have been adjusted.

Comments to Reviewer #3

In this manuscript, the authors developed a machine learning (ML) platform for retina OCT image segmentation and proposed a “gearbox-conceptualization” of ML to uncover the ML learning strategy in order to provide knowledge of its workflows. The results show that the differences between the platform and human graders were smaller than the total inter-grader variability.

Comment 1: Originality: The methodology presented in this work is not novel and based on the existing advances in machine learning.

Response 1: While we agree that the parts of the methods presented in this work may be not novel in themselves, we believe that the combination of these methods in the context presented has novelty. This is,

1. Having a group of three human experts in ophthalmologic image analysis generating a ground truth set of macaque OCT B-scans, which exhibits a certain level of ambiguity because there is variability among the human experts as on how they draw the labels.
2. Training a CNN on this ambiguous ground truth set.
3. Quantitatively comparing the CNN’s predictions on an independent test set, which was labelled by each of the three human graders, using the Hamming distance metric.
4. Visualizing the measured Hamming distances by using Multidimensional Scaling plots.

5. Discussing the different “averaging” tendencies of the CNN among the three human graders with respect to the four compartments (vitreous, retina, choroid, sclera) in the ground truth.
6. Finally, a search in PUBMED which comprises more than 30 million citations for biomedical literature) no reports were found using the keywords “machine learning, optical coherence tomography, hamming distance”. Using the keywords “image, machine learning, hamming distance”) only one (not relevant) publication was found (Local Multi-Grouped Binary Descriptor With Ring-Based Pooling Configuration and Optimization. Gao Y, Huang W, Qiao Y. IEEE Trans Image Process. 2015 Dec;24(12):4820-33. doi: 10.1109/TIP.2015.2469093. Epub 2015 Aug 17. PMID: 26292340 .

Due to these scarce results, it can be assumed that the presented combination has certainly has a novelty value and is worth to be shared with the scientific community. Therefore, we hope that the reviewer would reconsider and agree with the enhancements promoted by his valuable contribution to this manuscript.

Comment 2: The authors trained a deep CNN model which consisted of 22 convolutions, 5 transposed convolutions and 5 skip connections to segment OCT images automatically. However, the authors used only 8 and 32 eyes for testing and the ground truth generation for CNN algorithm. The OCT scan for each eye was exported to a stack of 25 B-scans. The dataset for this kind of learning-based method seems rather small, which makes it difficult to interpret the validation of results.

Response 2: We understand the reviewer’s concern regarding the size of the data set. Traditionally, ML needs a large amount of data. However, compared to other AI domains, in medicine patient derived data may be sparse. Nevertheless, we were able to show a quite good ML performance, which can be helpful and applicable especially for rare diseases with few patients, too:

To address the reviewer’s comment, we decided to increase the ground truth data set by 12.5% from 32 eyes to 36 eyes. In particular, each grader now contributed the same number of B-scans to the ground truth set. This is, 300 B-scans and 12 eyes. We agree with the reviewer that it’s important to include the same number of B-scans in the ground truth from each grader. Regarding the size of the ground truth data set: we performed initial experiments with 300 and 600 B-scans and we achieved overall accuracies of 98.0 and 98.4%, respectively, on the test set. We achieved an overall accuracy of 98.3% on the test set with the CNN described in this study, which was trained on the full 900 B-scans of the ground truth data set. Based on these results we believe that the size of 900 B-scans of the ground truth is sufficient to sustain the claims we make in this study. This high accuracy demonstrates that fewer data can be used than in traditional ML projects.

We have added this important comment in the discussion:

“Compared to other reports using CNNs a limit of this study could be the relatively low number of annotated ground truth data. However, the average Hamming distance between the human graders and the CNN was 0.0175 corresponding to 1.75% of pixels being labeled differently by the human graders and the CNN, respectively. This high predictive

performance of the CNN was confirmed when training on the smaller ground truth data set of 800 B-Scans, which yielded very similar results. This indicates that the ground truth size of 900 B-scans was sufficient to sustain the claims proposed in this study. However, it can be speculated that an even higher number of ground truth data could further improve the results. Nevertheless, the annotations of ground truth data by humans is a very time consuming process and the current study setup appears to be an acceptable compromise between human effort and CNN predictive performance, particularly considering that the development of ML algorithms often aims at reducing the human workload. “

Comment 3: The CNN algorithm learned from ground truth generated from three independent expert graders whereby each B-scan was labeled by one human grader. The results of CNN algorithm may be greatly affected by the human grader who labeled more B-scans. The authors need to add more labeling details by experts.

Response 3: We acknowledge the concern raised by the reviewer. Please see or Response 2. We increased the number of B-scans in the ground truth set contributed by all graders up to 300 images each. So that each of the three graders contributed the same number of B-scans (300 B-scans, 12 eyes) to the ground truth set. Also we have adjusted to manuscript to:

“To train and test the CNN, three retina experts manually graded”

Comment 4: In the "neural recording" method section, the authors made explanatory analysis of the model through the balanced performance for CNN results, without explaining the internal working mechanism of the model, which is not consistent with the "ML- gearbox" described in the manuscript.

Response 4: By the term neural recording (NR) we understand the recording and visualization of the predictive performance of CNN and human graders related to the ambiguity in the ground truth data, so that it is presented in a way that is best suited for a human cognitive scale. We have included this definition in the Method and discussion such as:

“Definition of neural recording

With regard to the overlap between neuroscience and ML, it can be argued that CNNs currently cannot function without a computer and their internal mechanisms are often opaque. The observation and possible recording of the activity of such computer-based circuits could be described in a simplified way as "neural recording". Since every deep learning model running on a computer already would fit that description, we propose a neural recording framework to better understand and visualize how the recorded predictive performance of the CNN relates to the ambiguity in the ground truth data. Ideally, such a neural recording is presented in a way that is appropriate for a human cognitive scale.”

Also, we have explained the term “gear box in more details” in the discussion:

“T-REX has three key components (in the following simply designated as “gears”): gear (1) ground truth generation by three human graders, gear (2) calculation of Hamming distances among all human graders and the machine’s predictions, and gear (3) a sophisticated data visualization thereof that we would like to describe as “neural recording.”

Comment 5: How do the authors propose to implement the AI system for real-world application.

Response 5: This is an interesting point. However, we have to note that we are at an early stage of the proposed T-REX concept. We have extended the discussion with regard to potential applications of T-REX real-world scenarios:

“More importantly, physicians will only make a clinical decision based on a recommendation of such an AI system if they can fully identify themselves with the AI. A subset of XAI methods aims at revealing post-hoc insights into “why” a machine has taken a certain decision. While well-known post-hoc approaches such as LRP or GradCAM visualize relevant regions in the input data, T-REX, our proposed XAI method, visualized and evaluated similarities between the CNN predictions and the labels of different humans that the CNN has learned from. Therefore, this study contributes to a better explainability in the application of AI, such that a resulting DL model can be better appreciated.

T-REX can provide a rigorous evaluation and re-calibration tool to incorporate new DL standards. In a more general sense, it can increase the quality of explanations that are based on DL systems, which increases causability. This in turn can promote safety for doctors and patients. Accordingly, the proposed post-hoc XAI approach T-REX is expected to enable data scientists to model more transparent DL systems. In return, this leads to more trust in trained DL models by physicians, which utilize DL for data-supported clinical decisions.”

Comment 6: The statistics used should be defined in the paper. The 95% confidence interval should be reported.

Response 6: We acknowledge that the presentation of the statistical test results was not clear enough in the previous version of this manuscript. This might have made the statistical analysis described in this study difficult to understand. However, we believe we described the statistical tests themselves in the last paragraph of the method section. This is, statistical permutation tests to compare the mean inter-grader Hamming distance among human graders with the mean Hamming distance between human graders and the CNN. We used permutation tests instead of t-tests because the distribution of the data was clearly in which case t-tests can produce unreliable results.

We decided to improve the wording in the presentation of the statistical test results. In addition, we think that providing confidence intervals, as suggested by the reviewer, is a good idea and increases the information content. We added 99% confidence interval (99% CIs were easily available from R). We reformulated the presentation of the statistical results in the result section to:

“The statistical permutation tests revealed that the mean human inter-grader Hamming distances were significantly larger than the mean Hamming distances between the humans and the CNN across all compartments and for the compartments vitreous, choroid and sclera separately. The recovered p-values were all $2e-5$ with a 99% confidence interval of $(0, 1e-4)$ for the p-values. This is, of all the permutations drawn, not a single time did the mean Hamming distance between humans and the CNN exceed the mean human inter-grader Hamming distance. On the other hand, for the retina compartment the mean inter-grader Hamming distance was significantly smaller than the mean Hamming distance between the

humans and the CNN. The recovered p-value was again $2e-5$ with a 99% confidence interval for the p-value of $(0, 1e-4)$."

Comment 7: Illustration of the OCT segmentations was missed in Figure 1.

Response 7: we have added a novel fig. 5 to illustrate the segmentation concept.

Comment 8: The manuscript needs to be polished.

Response 9: We hope that the current adjustments have contributed to a better readability. We are very thankful for the very good comments and clearly, the manuscript has gained much!

REVIEWERS' COMMENTS:

Reviewer #2 (Remarks to the Author):

This reviewer is of the opinion that the authors did a nice job and addressed all reviewers comments in an adequate manner, therefore this reviewer would now recommend acceptance of this paper. Congratulations to the authors!

Reviewer #3 (Remarks to the Author):

In this revised manuscript, the authors have revised the paper in line with reviewers' comments. The article applied a CNN for optical coherence tomography (OCT) image segmentation and T-REX to make the ML decisions more transparent. However, a number of issues remain.

1. Originality: Though the authors discussed contributions to the field of explainable AI (XAI). The methodology presented in this work is not novel and based on the existing advances in machine learning. All the methods (U-net model, hamming distance, MDS plots) are already well-known.
2. The work lacks of a ground-truth criterion. For example, for a 3-pixel 0-1 segmentation task, the hidden ground truth is $[1,1,1]$. Three graders classified it to $[0,0,1]$, $[0,1,0]$ and $[1,0,0]$ respectively, and the model classified it to $[0,0,0]$. Surely, using the hamming distance metric, the model learnt a 'robust' result which is the worst one. It cannot be implied that the approach can enhance people's trust to DL, as the authors mentioned in the discussion part.
3. The real-world application of T-REX is limited.

Basel, 12 December 2020

Dear editor and reviewers

Thank you very much for the good comments to enhance our manuscript entitled

“Unraveling the deep learning gearbox in optical coherence tomography image segmentation towards explainable artificial intelligence” COMMSBIO-20-1698A

We sincerely hope that we have addressed the concerns according to the best possible recommendations. At this opportunity, we would like to thank the reviewers, because they have greatly enriched this manuscript with their valuable comments.

Thank you so much for your kind consideration of our revised manuscript.

Peter Maloca

Reply to reviewer #3

Comment 1. Originality: Though the authors discussed contributions to the field of explainable AI (XAI). The methodology presented in this work is not novel and based on the existing advances in machine learning. All the methods (U-net model, hamming distance, MDS plots) are already well-known.

Reply 1 to reviewer 3:

Thank you very much for this comment. We have included and discussed this valuable comment in the limits and enhanced the manuscript.

The concept of originality varies considerably depending on the scientific discipline. In computer science, a new algorithm is considered a novel contribution. In domains such as computational biology, a new application of an algorithm to a certain task is regarded as a novel approach. In human-computer research, enhancing the understanding of human interactions with computers represents a novel feature. We believe our work is at the intersection of applied computer science, biomedical imaging, and explainable AI. We performed a literature search in Pubmed to

investigate how many publications with a topic and approach similar to our manuscript's are already available (<https://pubmed.ncbi.nlm.nih.gov/>, last visit 09 December, 2020): It turned out that among 30 million publications, not a single publication featured an approach and methodology comparable to ours, which we believe is an indication of a certain level of originality.

However, we consider it important to include the originality/novelty issue explicitly in the paper's discussion part and to stress that the methods that are part of T-REX are not novel in themselves but it's rather the novel combination of pre-existing concepts that makes up T-REX's originality.:

“The individual elements used in this manuscript (U-net model, Hamming distance, and MDS plots) do not appear to represent true methodological originality on their own. However, the concept of originality varies considerably depending on the scientific discipline. Our scientific originality was defined as a permutation of new and old information¹ and the unique arrangement of existing elements of ML techniques for the application in computational biology². The scientific discoveries of this study thus provide subsequent studies with a unique combination of knowledge not available from previous reports. In short, the appropriate conceptualisation of the mentioned ML techniques into the proposed frame-work, improved the understanding of the interface between automatic computing and life sciences and therefore represents nevertheless a specific degree of originality.

“It should be noted that the individual elements used in this study, i.e. U-Net, Hamming distance, MDS and heatmap plots, do not represent methodological novelty on their own. The scientific originality of our work, however, can be viewed as a unique combination of pre-existing components² or as a permutation of new and old information¹. T-REX and its associated scientific discoveries in this study provide subsequent studies with a novel methodology and a combination of knowledge not available from previous reports. In short, the appropriate conceptualisation of the mentioned ML elements into the proposed framework improved the understanding of the interface between automatic computing and life sciences and therefore represents nevertheless a specific degree of originality.”

1 Dirk, L. A Measure of Originality: The Elements of Science. Social Studies of Science 29, 765-776, doi:10.1177/030631299029005004 (1999).

2 Shibayama, S. & Wang, J. Measuring originality in science. Scientometrics 122, 409-427, doi:10.1007/s11192-019-03263-0 (2020).

2. Reviewers comment: ground-truth criterion The work lacks of a ground-truth criterion. For example, for a 3-pixel 0-1 segmentation task, the hidden ground truth is [1,1,1]. Three graders classified it to [0,0,1], [0,1,0] and [1,0,0] respectively, and the model classified it to [0,0,0]. Surely, using the hamming distance metric, the model learnt a 'robust' result which is the worst one. It cannot be implied that the approach can enhance people's **trust** to DL, as the authors mentioned in the discussion part.

Reply 2 to reviewer:

We understand the comment of the reviewer. Indeed, our work does not include a ground truth criterion. Such a ground truth criterion would require an unambiguously correct or "true" ground truth - the reviewer uses the expression "hidden ground truth". In our application, however, we believe there is no such unambiguously correct ground truth. This is, because OCT experts don't completely agree on the manual segmentation of OCT images into four compartments (vitreous, retina, choroid, sclera). This difference in annotation does not arise because of random deviations from an optimal segmentation. Rather these differences arise because there are small but systematic differences in where the experts believe the compartment borders to be in the OCT scans. And there is no straight-forward way to eliminate this ambiguity. Usually, we can't dissect an eye after an OCT scan was recorded to find out where the physical compartment border really is.

We thus consider the ground truth generated by each of the three human experts equally true and investigate how our CNN reacts to this ambiguity in the ground truth. If indeed the three graders would annotate a 3-pixel 0-1 segmentation task as [0,0,1], [0,1,0], and [1,0,0], respectively, we would consider each of these annotations equally true. In our setting, we lack the possibility to define [1,1,1] as the unambiguously true labels. Furthermore, we think that this unresolvable ground truth ambiguity is a common phenomenon in medical imaging tasks. Thus, our contribution of investigating in how a machine learning algorithm reacts to ambiguous ground truth is especially insightful. Finally, the use of ambiguous ground truth as baseline helps to reduce bias in trained ML models.

However, we acknowledge that the lack of a of an unambiguous ground truth criterion is an important point which we didn't highlight enough. We thank the reviewer for pointing it out and we decided to address the unambiguous-ground-truth-criterion explicitly in the Introduction as:

“Supervised machine learning requires labeled ground truth data, which is usually annotated by humans. However, in some domains there is no absolute consensus on what the true ground truth labels should be. This is particularly true in medical imaging. Different experts might judge the same medical images slightly differently and come to different conclusions as to where e.g. the borders of certain medical structures should be²⁴. In some cases, the ambiguity might be irresolvable, i.e. there is no unambiguous ground truth criterion, since it’s impossible to determine the exact location of these structures without applying invasive procedures to the patient. In these cases, we might want to understand how a ML algorithm reacts to ambiguity in the ground truth data. To shed light on this issue, we trained a CNN from ambiguous ground truth consisting of labels from three human graders who acted as three CNN “teachers”.”

Besides, we acknowledge the reviewer’s point that it’s too strong a statement to say, “our approach can enhance people’s trust to DL”. We thus reformulated “trust” to “further insights”.

3. The real-world application of T-REX is limited.

Reply 3 to reviewer3:

We highly appreciate your comment! Thank you!

The aim of our work was to show the relationship between an ambiguous human-generated ground truth and the predictions of a CNN that learned from it using a visual display method that is quite intuitive not only for AI experts but also for average researchers. We believe the real-world application of T-REX is not as limited as it might seem at first. Ambiguity or uncertainty in ground truth data is probably a quite common phenomena in machine learning applications. Be it because of true/irresolvable ambiguity, like with the medical data set used in our study, or be it because humans make annotation mistakes. Requiring all human graders to annotate the same “test set” allows quantifying both, the inter-grader agreement, and the agreement between human graders and the machine learning algorithm that learned from the human data.

However, we acknowledge that the discussion of real-world application of T-REX was not sufficient in the manuscript and that these applications can’t be obvious for a reader. Therefore, we expanded the last paragraph of the discussion including an explicit discussion of real-world applications:

“The proposed method T-REX is not limited to semantic image segmentation in ophthalmology. In fact, it can be applied to improve the understanding of any machine learning algorithm that learns from ambiguous ground truth data. For example, T-REX could be used in the application of uncovering biases of ML prediction models in digital histopathology not only with respect to dataset biases, but also with respect to varying opinions of experts labeling the histopathology images.³ In applications, where supervised ML decision models are trained to detect new diseases such as Covid-19⁴ and experts still need to explore and agree upon specificities of the new disease, T-REX would be helpful to visualize the ambiguity of the experts’ opinions, i.e., labels. Hence, T-REX might be especially important if the ambiguity is irresolvable meaning that domain experts disagree about the true labels, but the differences cannot be eliminated in a straightforward way. In many medical applications, the true labels cannot be verified because applying invasive procedures to patients is impossible. Therefore, methods such as T-REX, which highlight the results of the model training from ambiguous ground truth, help to improve the understanding of the objectivity of a trained model and reduce bias in the ground truth. In a wider context, T-REX might yield insights into how AI algorithms make decision under uncertainty, a process very familiar to humans but less understood in the field of AI. “

³ Resolving challenges in deep learning-based analyses of histopathological images using explanation methods, 2020, Hägele et al. <https://www.nature.com/articles/s41598-020-62724-2.pdf?origin=ppub>

⁴ A deep learning and grad-CAM based color visualization approach for fast detection of COVID-19 cases using chest X-ray and CT-Scan images, 2020, Panwar et al, <https://www.sciencedirect.com/science/article/pii/S0960077920305865?via%3Dihub>